# Common mechanism of transcription termination at coding and noncoding RNA genes in fission yeast

Marc Larochelle[1], Marc-Antoine Robert[2], Jean-Nicolas Hébert[1], Xiaochuan Liu[3], Dominick Matteau[2], Sébastien Rodrigue[2], Bin Tian[3], Pierre-Étienne Jacques[2,4] & François Bachand[1,4]

Termination of RNA polymerase II (RNAPII) transcription is a fundamental step of gene expression that is critical for determining the borders between genes. In budding yeast, termination at protein-coding genes is initiated by the cleavage/polyadenylation machinery, whereas termination of most noncoding RNA (ncRNA) genes occurs via the Nrd1–Nab3–Sen1 (NNS) pathway. Here, we find that NNS-like transcription termination is not conserved in fission yeast. Rather, genome-wide analyses show global recruitment of mRNA 3′ end processing factors at the end of ncRNA genes, including snoRNAs and snRNAs, and that this recruitment coincides with high levels of Ser2 and Tyr1 phosphorylation on the RNAPII C-terminal domain. We also find that termination of mRNA and ncRNA transcription requires the conserved Ysh1/CPSF-73 and Dhp1/XRN2 nucleases, supporting widespread cleavage-dependent transcription termination in fission yeast. Our findings thus reveal that a common mode of transcription termination can produce functionally and structurally distinct types of polyadenylated and non-polyadenylated RNAs.

[1] Département de Biochimie, Université de Sherbrooke, Sherbrooke, QC J1E4K8, Canada. [2] Département de Biologie, Université de Sherbrooke, Sherbrooke, QC J1K2R1, Canada. [3] Department of Microbiology, Biochemistry and Molecular Genetics, Rutgers New Jersey Medical School and Rutgers Cancer Institute of New Jersey, Newark, NJ 07103, USA. [4] Centre de Recherche du CHUS, Université de Sherbrooke, Sherbrooke, QC J1H5N4, Canada. These authors contributed equally: Marc Larochelle, Marc-Antoine Robert. Correspondence and requests for materials should be addressed to P.-É.J. (email: Pierre-Etienne.Jacques@USherbrooke.ca) or to F.B. (email: f.bachand@usherbrooke.ca)

RNA polymerase II (RNAPII) is responsible for the synthesis of a broad range of coding and noncoding transcripts, and termination pathways have a decisive influence on the fate of transcribed RNAs[1]. Although transcription termination has been investigated in a wide range of organisms, it has been most extensively studied in the model organism *Saccharomyces cerevisiae*, where two major termination pathways exist depending on the class of genes transcribed[2]. For protein-coding genes, data support a mechanism triggered by the cleavage activity of the Ysh1 endonuclease (CPSF-73 in humans), which is part of a nuclease module within the larger mRNA 3′ end processing complex[3]. Specifically, the co-transcriptional recruitment of conserved cleavage and polyadenylation factors at poly(A) signal (PAS) causes cleavage of the nascent pre-mRNA, followed by 3′ end polyadenylation of the released transcript by the poly(A) polymerase[4]. This cleavage also provides a free and uncapped 5′ entry point for a protein complex that includes an evolutionarily conserved 5′–3′ exonuclease (XRN2 in humans, Rat1 in *S. cerevisiae*; Dhp1 in *S. pombe*). The exonuclease is thought to chase RNAPII and promote its dissociation from the DNA template[5–8], a mechanism referred as "torpedo"-mediated transcription termination.

In addition to mRNAs, RNAPII synthesizes an extensive set of noncoding RNAs (ncRNAs) that include small nucleolar RNAs (snoRNAs), small nuclear RNAs (snRNAs), long noncoding RNAs (lncRNAs), and cryptic unstable transcripts (CUTs). In *S. cerevisiae*, transcription of snoRNAs, snRNAs, and CUTs does not rely on the cleavage/polyadenylation machinery for termination, but instead uses a cleavage-independent termination pathway that requires a complex (referred to as NNS complex) consisting of the RNA-binding proteins Nab3 and Nrd1 as well as the DNA/RNA helicase Sen1[2]. In this mode of termination, the NNS complex interacts with both the transcription machinery[9,10] and specific RNA motifs enriched downstream of *ncRNA* genes[11,12] to engage the transcription elongation complex via the Sen1 helicase, which translocate onto the nascent RNA and catches up with the transcribing polymerase to elicit termination[13].

One transcription-associated feature that appears to influence the choice between torpedo- and NNS-mediated termination is the phosphorylation status of the carboxy-terminal domain (CTD) of the RNAPII catalytic subunit, Rpb1. The CTD consists of a succession of conserved heptad repeats, with the consensus amino acid sequence Y-S-P-T-S-P-S[14]. The RNAPII CTD is subjected to a plethora of post-translational modifications throughout the transcription cycle that are key to coordinate the sequential recruitment of RNA-processing factors. Among CTD modifications, phosphorylation of Ser2 and Ser5 have been most extensively studied and appear to be the most abundant in *S. cerevisiae*[15]. In the cases of small ncRNA genes, NNS recruitment is influenced by Ser5 phosphorylation (Ser5-P) via the CTD interaction domain (CID) of Nrd1[9,10]. As transcription elongation progresses into protein-coding genes, the ratio between the levels of phosphorylated Ser2 (Ser2-P) and Ser5-P gradually increases, peaking near the mRNA cleavage site[16,17]. Pcf11, a component of the mRNA 3′ end processing machinery, preferentially recognizes Ser2-P CTD repeats via its CID domain[18,19], which may explain the robust levels of Pcf11 at the 3′ end of protein-coding genes[20]. Yet, Pcf11 recruitment at the 3′ end of mRNA genes may also be influenced by additional CTD modifications. For instance, Tyr1 phosphorylation (Tyr1-P) can impair binding of Pcf11 to Ser2-P CTD peptides in vitro[20]. Accordingly, it was proposed that Tyr1-P prevents recruitment of termination factors in the coding region of genes in *S. cerevisiae*, and that timely dephosphorylation of Tyr1-P upstream of the polyadenylation site by the Glc7 phosphatase would allow for recruitment of the 3′ end processing machinery via recognition of Ser2-P CTD repeats by the CID domain of Pcf11[21].

High-throughput sequencing analyses indicate that the majority of genomes are transcriptionally active, thus yielding a large amount of ncRNAs[22,23]. Yet, how RNAPII transcription is terminated at ncRNA genes remains poorly understood for many eukaryotic species. Also unclear is whether the use of distinct pathways to terminate coding and noncoding transcription, such as described in *S. cerevisiae*, is evolutionarily conserved. Recently, we have shown that the *S. pombe* homolog of *S. cerevisiae* Nrd1, Seb1, does not physically associate with Nab3 and Sen1 homologs in fission yeast[24]. Here we show that the absence of Nab3 and Sen1 homologs in *S. pombe* does not affect transcription termination of coding and ncRNA genes. Unexpectedly, we find that mRNA 3′ end processing factors are enriched at the 3′ end of ncRNA genes and that this recruitment coincides with high levels of Ser2 and Tyr1 CTD phosphorylation. Consistently, we find that most independently transcribed fission yeast snoRNA genes are cleaved and polyadenylated in a manner dependent on conserved 3′ end processing factors, and that termination at ncRNA genes is sensitive to deficiencies in Ysh1 and Dhp1. Our findings indicate that torpedo-mediated transcription termination is widespread in fission yeast, thereby revealing that a universal mode of transcription termination can promote the synthesis and accumulation of both polyadenylated mRNAs and non-polyadenylated ncRNAs.

## Results

**Nab3 and Sen1 are not required for RNAPII termination**. To measure the global impact of mutations in NNS components on RNAPII transcription in *S. pombe*, we analyzed genome-wide RNAPII (Rpb1) occupancy by ChIP-seq using *nab3Δ*, *sen1Δ*, and *dbl8Δ* (Sen1 paralog) single mutants as well as a *sen1Δ dbl8Δ* double mutant. The absence of Nab3, Sen1, and Dbl8 did not result in noticeable read-through transcription at mRNA and snoRNA genes compared to the wild-type (WT) control. In contrast, a deficiency in the essential protein Seb1 results in defective termination at most RNAPII-transcribed genes (see *Pnmt-seb1* mutant; Fig. 1a, b)[24]. Given that Nab3, Sen1, and Dbl8 do not copurify with Seb1[24,25], and considering the absence of termination defects in *nab3Δ, dbl8Δ, sen1Δ*, and *sen1Δ dbl8Δ* mutants (Fig. 1a, b), we conclude that NNS-dependent transcription termination is not conserved in fission yeast.

**Recruitment of mRNA 3′ end processing factors at ncRNA genes**. The lack of conservation of NNS-dependent transcription termination raised the question as to the mechanism of 3′ end processing and transcription termination at ncRNA genes in fission yeast. One clue about how ncRNAs could be processed in *S. pombe* derives from studies on the nuclear poly(A)-binding protein Pab2, the homolog of human PABPN1[26]. We previously uncovered a polyadenylation-dependent pathway required for the maturation of independently transcribed snoRNAs that relies on Pab2[27,28]. To test for the possibility that mRNA cleavage and polyadenylation factors function in 3′ end processing of ncRNA genes in *S. pombe*, we used ChIP-seq to examine the genome-wide-binding profile of proteins with conserved roles in mRNA 3′ end processing and transcription termination. This analysis included two independent components of the cleavage and polyadenylation factor I (CFI) complex, namely Rna14 and Pcf11, the endonuclease Ysh1 that is responsible for pre-mRNA cleavage before 3′ end polyadenylation, and Dhp1, the homolog of *S. cerevisiae* Rat1 and human XRN2 that promote transcription termination of RNAPII via 5′–3′ exonucleolytic activity. The ChIP-seq profile of Seb1[24] was also included, as this protein was shown to be important for 3′ end processing of both mRNAs and ncRNAs. Consistent with roles in 3′ end processing at mRNA

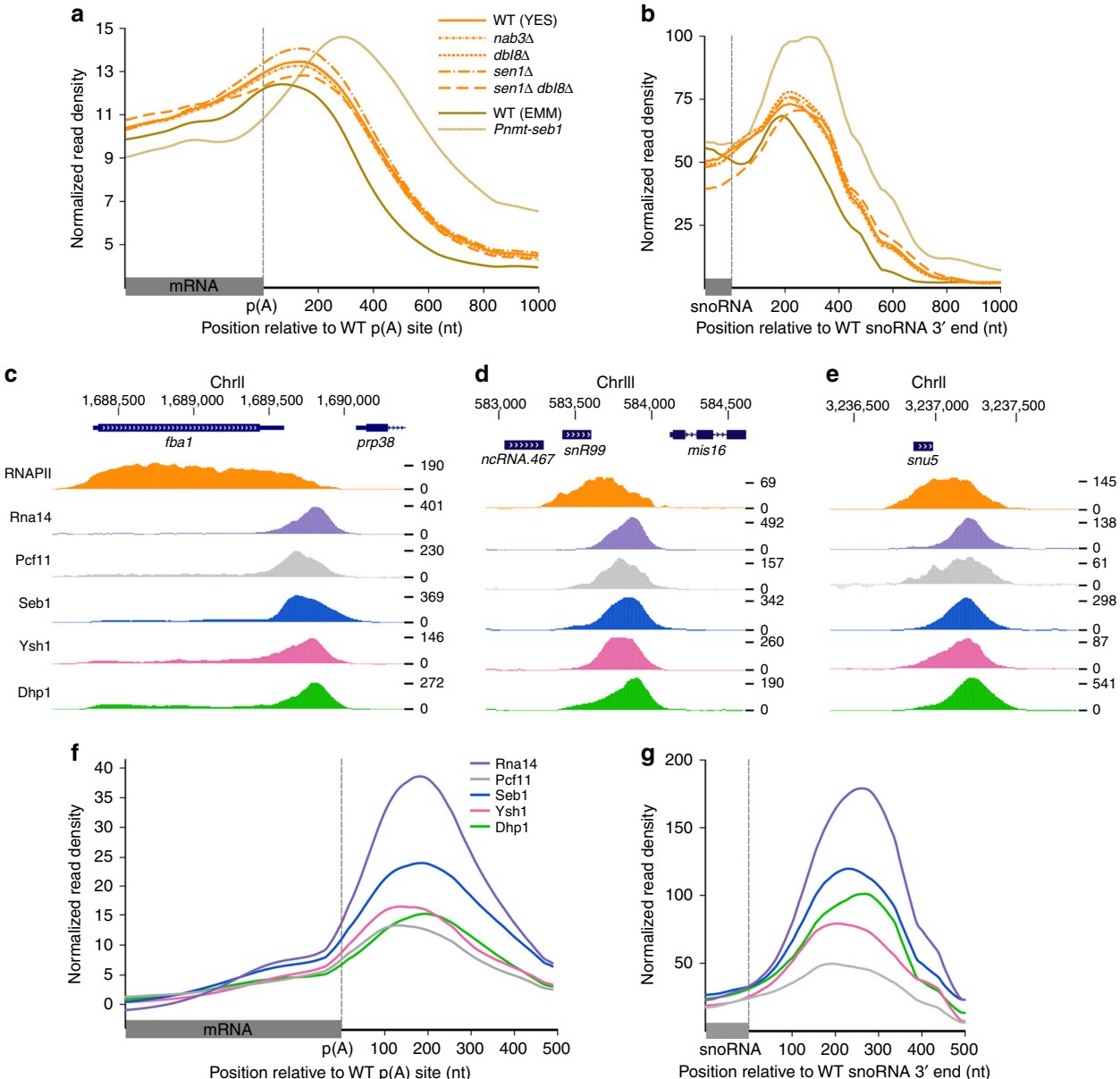

**Fig. 1** *S. pombe* mRNA 3′ end processing and transcription termination factors are recruited at the 3′ end of independently transcribed snoRNA and snRNA genes. **a**, **b** Average ChIP-seq profiles of total RNAPII (Rpb1) relative to mRNA poly(A) site (**a** n = 4755) and to annotated 3′ end of independently transcribed monocistronic snoRNAs (**b** n = 31) in WT (solid line, orange) and NNS mutant strains (dotted lines, orange) grown in rich medium (YES). Average ChIP-seq profiles of total RNAPII (Rpb1) in WT and Seb1-deficient cells (*Pnmt-seb1*) grown for 15 h in thiamine-supplemented minimal medium (EMM) are also shown (gold). Gene coordinates are available in Supplementary Data 1–2. **c–e** Normalized ChIP-seq signal of RNAPII (Rpb1) and the indicated mRNA 3′ end processing factors across the *fba1* mRNA (**c**), the *snR99* snoRNA (**d**), and the *snu5* snRNA (**e**) genes. **f**, **g** Average ChIP-seq profile of the indicated mRNA 3′ end processing factors over the same groups of mRNA (**f**) and snoRNA (**g**) genes as **a**, **b**

genes, Rna14, Pcf11, Ysh1, Seb1, and Dhp1 all showed strong binding at the 3′ end of protein-coding genes (Fig. 1c, f). Notably, they also exhibited robust enrichment at the 3′ end of independently transcribed snoRNAs (Fig. 1d, g) and snRNA (Fig. 1e) genes. In contrast, these factors were not enriched at the 3′ end of intron-encoded snoRNAs (Supplementary Fig. 1), consistent with a maturation pathway independent of cleavage and polyadenylation. Interestingly, analysis of peak distribution based on average-binding profiles indicated that the recruitment of Dhp1 generally occurred downstream of the endonuclease Ysh1 (Fig. 1f, g; note that the peak of the green curves is shifted downstream relative to the pink curves). This is consistent with transcription termination of coding and ncRNA genes following the torpedo

model, in which Dhp1 requires Ysh1-dependent endonucleolytic cleavage to promote disengagement of RNAPII from the DNA template[5–7]. On the basis of these results, we conclude that mRNA 3′ end processing factors are recruited at the 3′ end of both coding and ncRNA genes in fission yeast.

**Synthesis of ncRNAs requires mRNA 3′ end processing factors.** The presence of cleavage and polyadenylation factors at the 3′ end of ncRNA genes suggested a role for the mRNA 3′ end processing machinery in snoRNA synthesis. To examine the requirement of Rna14, Pcf11, Ysh1, and Dhp1 in the synthesis of ncRNAs, conditional strains were generated since these proteins

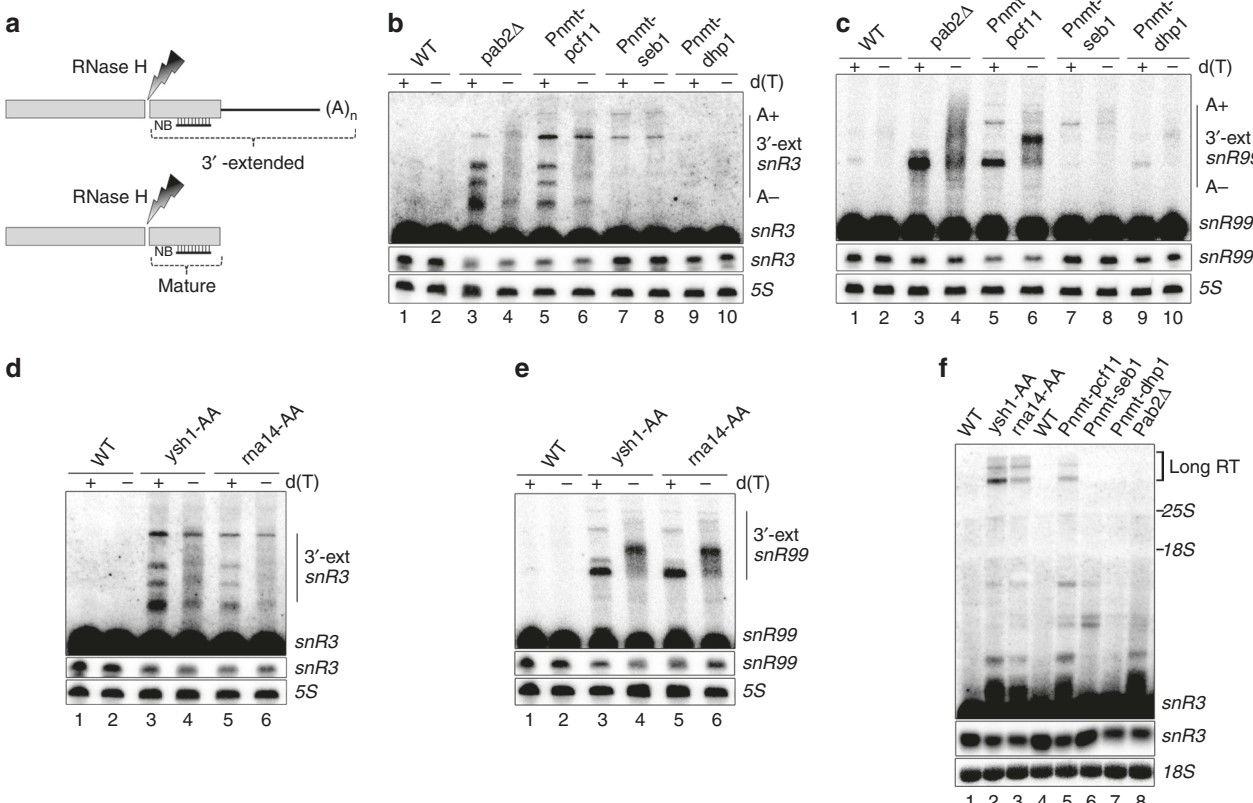

**Fig. 2** The mRNA cleavage and polyadenylation complex is required for snoRNA synthesis. **a** Schematic of the RNase H cleavage assay used in **b**–**e**. After RNase H cleavage of the snoRNA at sites of RNA:DNA hybrids in the presence of a sequence-specific DNA oligonucleotide, the 3′ fragment (mature or 3′-extended) is detected by Northern blotting (NB). Addition of oligo d(T) to the RNase H reaction removes heterogenous poly(A) tails, generating discrete products. **b**, **c** Total RNA prepared from the indicated strains that were previously grown in thiamine-supplemented medium for 15 h to deplete Pcf11, Seb1, and Dhp1 was treated with RNase H in the presence of DNA oligonucleotides complementary to snR3 (**b**) and snR99 (**c**). RNase H reactions were performed in the presence (+) or absence (−) of oligo d(T). The top panel represents a longer exposure of the middle panel to see 3′-extended (3′-ext) cleavage products. The 5S rRNA was used as a loading control. **d**, **e** As described in **b**, **c**, but using cells that were previously treated with rapamycin to deplete Ysh1 and Rna14 from the nucleus. **f** Northern blot analysis using total RNA prepared from the indicated strains that were treated with either rapamycin (lanes 1–3) or thiamine (lanes 4–8). The blot was hybridized using DNA probes specific to snR3 and the 18S rRNA. The position of mature snR3, 18S and 25S rRNAs, as well as snR3 read-through (RT) products is indicated on the right

are all encoded by essential genes. As used previously to repress the expression of essential core exosome subunits[29] and Seb1[24], we constructed strains with endogenous *pcf11* and *dhp1* genes under the control of the thiamine-repressible *nmt81* promoter (*Pnmt*; see Methods for details). Importantly, levels of Seb1, Pcf11, and Dhp1 expressed from the *nmt* promoter were similar to levels expressed from their endogenous promoter in non-repressing conditions, but showed robust depletion in thiamine-supplemented conditions when expressed under the *nmt* promoter (Supplementary Fig. 2a–c). Consistent with Pcf11 and Dhp1 being essential for viability[30], *Pnmt-pcf11* and *Pnmt-dhp1* cells cultured in thiamine-supplemented medium showed growth arrest (Supplementary Fig. 2d). Depletion of Pcf11 in *S. pombe* also affects mRNA production[31], as expected of a protein essential for mRNA synthesis. To generate conditional strains for *rna14* and *ysh1*, we used the rapamycin-dependent anchor-away system[32] (Supplementary Fig. 2e), as the *nmt* promoter did not provide sufficient Rna14 and Ysh1 depletion to induce growth arrest. Importantly, inactivation of Ysh1 and Rna14 by nuclear depletion (Supplementary Fig. 2f, g) impaired mRNA synthesis in a rapamycin-dependent manner (Supplementary Fig. 2h, i) and resulted in the production of read-through transcripts (Supplementary Fig. 2j), consistent with 3′ end processing defects at protein-coding genes. To assess the contribution of Pcf11, Dhp1,

Ysh1, and Rna14 to snoRNA 3′ end processing, we used an RNase H cleavage assay that can simultaneously detect both the mature snoRNA and 3′-extended polyadenylated snoRNA precursors (Fig. 2a). As a control, we used the *pab2Δ* mutant that accumulates 3′-extended polyadenylated precursors, resulting in 45 and 34% reduction in snR3 and snR99 snoRNAs levels[28] (Fig. 2b, c, compare lanes 3–4 to 1–2). Similarly, deficiencies in Pcf11, Ysh1, and Rna14 resulted in reduced levels of mature snR3 and snR99 snoRNAs: 39 and 42% decrease, respectively, for Pcf11 (Fig. 2b, c, lanes 5–6; Supplementary Fig. 2k); 45 and 46% decrease, respectively, for Ysh1 (Fig. 2d, e, lanes 3–4); and 41 and 33% decrease, respectively, for Rna14 (Fig. 2d, e, lanes 5–6). In addition to polyadenylated pre-snoRNA detected by RNase H assays (Fig. 2b, e), long snR3 read-through products were also detected by Northern blot analysis in cells deficient for Pcf11, Ysh1, and Rna14 (Fig. 2f, lanes 2–3 and 5). In contrast, snoRNA 3′-extensions did not accumulate in cells deficient for Dhp1 (Fig. 2b, c, lanes 9–10; Supplementary Fig. 2k; and Fig. 2f, lane 7), consistent with a role in transcription termination that occurs beyond 3′ end processing. As shown previously[24], a deficiency in Seb1 did not negatively impair RNA production as demonstrated by the similar levels of mature snoRNA between wild-type and Seb1-deficient cells, but changed cleavage site selection as observed by the lengthening of the 3′-extended snoRNA precursors on RNase

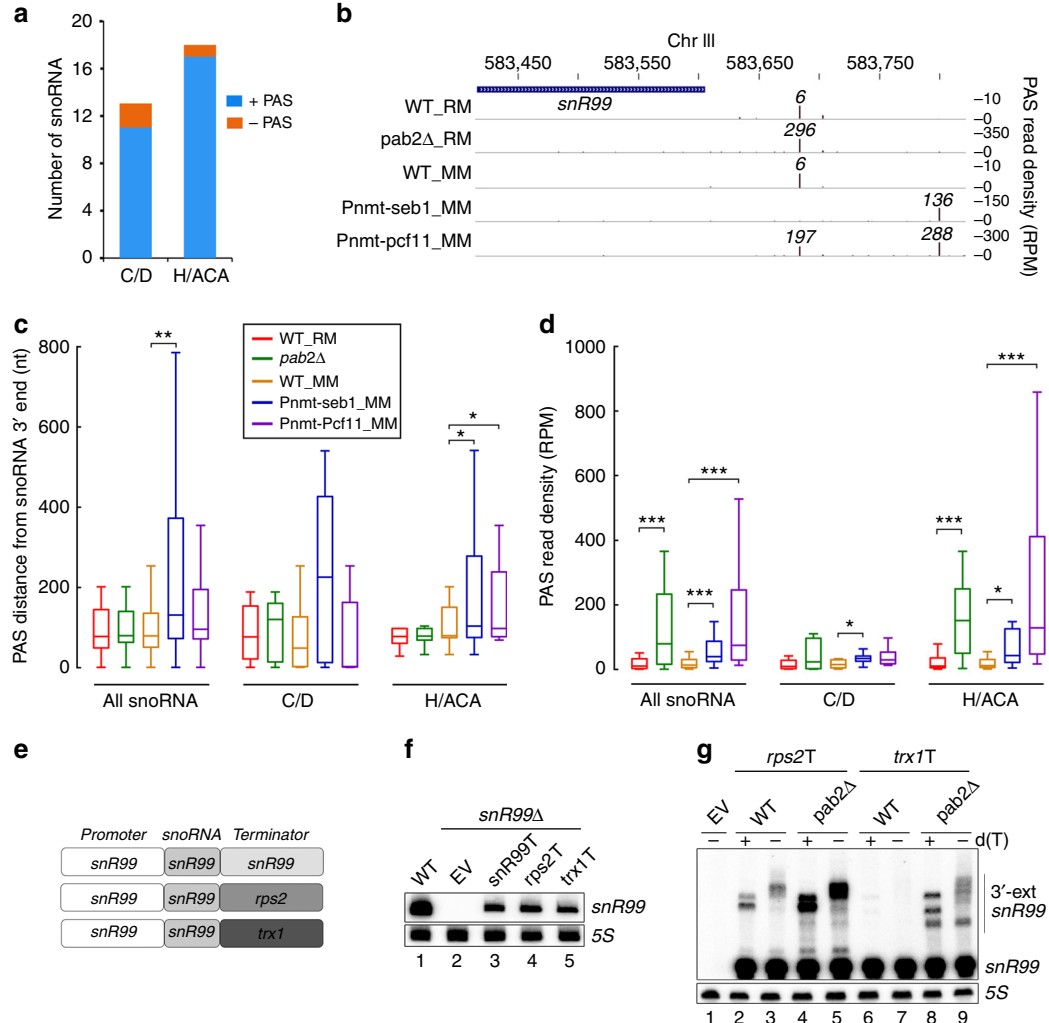

**Fig. 3** Independently transcribed snoRNA genes are cleaved and polyadenylated. **a** Proportion of 13 C/D box and 18 H/ACA box monocistronic snoRNAs with at least one poly(A) site mapped by 3′READS in the WT strain grown in minimal or rich media. **b** Poly(A) site (PAS) read density (in RPM) profile downstream of the snR99 snoRNA as determined by 3′ READS in the indicated strains grown in either rich (RM) or minimal (MM) media supplemented with thiamine for 15 h to deplete Seb1 and Pcf11. **c**, **d** Box plots showing the distribution of the distance calculated between the strongest poly(A) site (PAS) and the annotated snoRNA 3′ end (**c**), as well as the sum of the read density for all of the poly(A) sites associated to a snoRNA in each condition (**d**). Center lines correspond to the median and the significance of difference (Wilcoxon signed-rank test) is indicated for comparing groups (*$p < 0.05$; **$p < 0.01$; ***$p < 0.005$). **e** Schematic of snR99 constructs with different 3′ flanking sequences used for experiments in **f** and **g**. **f** Northern blot analysis of total RNA prepared from WT (lane 1) and snR99-null (lanes 2–5) cells that were previously transformed with the indicated constructs (EV, empty vector). **g** Total RNA prepared from snR99-null cells that were previously transformed with snR99 constructs that comprised rps2 (rps2T, lanes 2–5) or trx1 (trx1T, lanes 6–9) terminator sequences was treated with RNase H in the presence of DNA oligonucleotides complementary to snR99

H assays (Fig. 2b, c, compare lanes 1–2 and 7–8; Supplementary Fig. 2k). These results indicate that mRNA 3′ end processing factors are required for the synthesis of independently transcribed snoRNAs. Together with data showing recruitment of 3′ end processing factors downstream of ncRNA genes (Fig. 1), these results suggest that ncRNA 3′ end formation involves cleavage and polyadenylation.

**Widespread polyadenylation at snoRNA genes**. To obtain a comprehensive view of pre-snoRNA polyadenylation and to address whether differences exist between 3′ end processing of H/ACA and C/D box snoRNAs, we used 3′ region extraction and deep sequencing (3′READS), an approach developed to map mRNA polyadenylation sites at the genome-wide level[33]. Over 6 millions reads mapping to polyadenylation sites (PAS reads) were obtained from wild-type *S. pombe* cells grown in rich and minimal media by 3′ READS[34]. Although the majority (~60%) of unique PAS reads

mapped to 3′ UTR of protein-coding genes, roughly 18% of unique PAS locations mapped to intergenic regions. Notably, out of 31 independently transcribed monocistronic snoRNAs, 24 (77%) and 27 (87%) had a mappable poly(A) site located downstream of a snoRNA annotation in minimal and rich media, respectively (Supplementary Data 2). It should be noted that snoRNA genes for which PAS reads could not be identified showed barely detectable levels of RNAPII as determined by ChIP-seq assays. Comparison between C/D and H/ACA box snoRNAs indicated similar proportions of polyadenylated species (Fig. 3a). Figure 3b shows 3′READS data for the H/ACA box snR99 snoRNA. As can be seen, 3′READS identified a major PAS located 78-nt downstream of the annotated snR99 mature 3′ end in wild-type cells. In the pab2Δ mutant, 3′READS detected a >45-fold increase in the abundance of snR99 precursors that used this major PAS (Fig. 3b). Notably, whereas depletion of Seb1 resulted in the use of a single distal PAS located 118-nt downstream from the major snR99 PAS (196-nt from the snR99 mature 3′ end), the two different

PASs identified in the WT, *pab2Δ*, and *Pnmt-seb1* strains were used in cells deficient for Pcf11 (Fig. 3b). Collectively, the 3′READS data are entirely consistent with results obtained using RNase H cleavage assays (see Fig. 2c).

Global analysis of 3′READS data revealed that the median distance between the annotated (mature) snoRNA 3′ end and the pre-snoRNA cleavage site is 77-nt in wild-type cells grown in rich medium (Fig. 3c, red boxes). Consistent with previous results[28] (Fig. 2), the absence of Pab2 did not generally affect cleavage site selection at snoRNA genes (Fig. 3c, green boxes), but resulted in a significant accumulation of polyadenylated snoRNA precursors (Fig. 3d, compare green and red boxes); interestingly, H/ACA box pre-snoRNAs appeared to accumulate to greater levels as compared to C/D box precursors in the absence of Pab2 (Fig. 3d). In contrast, the absence of Pab2 did not generally impact the levels of polyadenylated mRNAs (Supplementary Fig. 3). In the case of Seb1, as seen for *snR99* (Fig. 3b), we noted a clear shift of the general distribution of polyadenylated pre-snoRNAs toward longer 3′-extensions (Fig. 3c, blue boxes): the median distance between the snoRNA mature 3′ end and its corresponding poly(A) site increased from 79-nt in wild-type cells grown in minimal medium to 131-nt in Seb1-deficient cells (compare blue and yellow boxes). This result is consistent with the preferential use of distal cleavage sites in the *seb1* mutant (Fig. 2), an outcome also observed for mRNAs[24] (Supplementary Fig. 3). The overall distribution of pre-snoRNA polyadenylation sites was also extended in Pcf11-deficient cells, a result that was greater for H/ACA box snoRNAs (Fig. 3c, compare purple and yellow boxes), as well as for mRNAs[34]. In addition, a general accumulation of polyadenylated pre-snoRNAs was observed in *pcf11* and *seb1* mutants (Fig. 3d, compare purple and blue boxes to yellow boxes). In summary, most independently transcribed snoRNA genes produce 3′-extended polyadenylated precursors in *S. pombe*.

A mechanism of snoRNA 3′ end processing that involves cleavage and polyadenylation by mRNA maturation factors posits that *cis*-acting elements present in the 3′ flanking region of coding and ncRNA genes should be similar. We and others have shown that poly(A) signals around *S. pombe* mRNAs are associated with upstream AAU[A/G]AA hexamers and downstream GUA motifs[34–36]. However, the relatively small number of independently transcribed snoRNAs ($n = 31$), as compared to mRNAs ($n = 4755$), limited the identification of significantly enriched sequence motifs around snoRNA poly(A) sites. We therefore tested whether *cis* elements promoting mRNA 3′ end processing could support 3′ end maturation of a snoRNA. For this, we generated constructs in which 1-kb of *snR99* downstream sequence was exchanged with 1-kb of downstream sequence from *rps2* and *trx1* protein-coding genes (Fig. 3e). These three constructs as well as a vector control were chromosomally integrated as a single copy into a *snR99*-null strain. As shown in Fig. 3f, 3′ flanking sequences from both *rps2* and *trx1* supported proper 3′ end formation and normal *snR99* accumulation, as demonstrated by the length and expression level of the snoRNA, which were similar to *snR99* expressed from its authentic downstream sequence (compare lanes 3–5). Furthermore, RNase H cleavage assays and 3′ RACE experiments revealed Pab2-dependent polyadenylated snoRNA precursors that used the natural *rps2* and *trx1* cleavage sites (Fig. 3g and Supplementary Fig. 3c, d). Based on these findings, we conclude that snoRNA 3′ end processing in *S. pombe* occurs via a pathway analogous to mRNA biogenesis.

**Tyr1-P and Ser2-P peak at the end of mRNA and ncRNA genes.** A number of studies have examined the association between the status of CTD phosphorylation and 3′ end processing/transcription termination. Since little is known about the association between CTD modifications and RNA processing in fission yeast, we mapped Tyr1, Ser2, Ser5, and Ser7 phosphorylation at the genome-wide level in wild-type *S. pombe*. ChIP-seq data from independent biological replicates showed high correlation throughout the *S. pombe* genome (see Supplementary Fig. 4). Averaging ChIP-seq signal across protein-coding genes revealed CTD phosphorylation patterns similar to those characterized in *S. cerevisiae* (Fig. 4a): Ser5 and Ser7 phosphorylation (Ser5-P and Ser7-P) marks peaking at the 5′ end of coding genes after the transcription start site (TSS), whereas Ser2 phosphorylation (Ser2-P) steadily increased along the gene body and sharply peaked downstream of the poly(A) site. The pattern of Tyr1 CTD phosphorylation (Tyr1-P) at protein-coding genes was similar to that of Ser2-P, peaking sharply downstream of the poly (A) site (Fig. 4a). It is interesting to note that the genome-wide pattern of Tyr1-P signal in *S. pombe* contrasts to that of *S. cerevisiae* in which Tyr1-P signal was shown to peak upstream of the poly(A) site[20]. Occupancy profiles of CTD phosphorylation marks at snoRNA genes showed a pattern somewhat similar to that of mRNA genes, with peaking Ser5-P and Ser7-P signals at the 5′ end that started decreasing in the snoRNA gene body (Fig. 4b). As seen for mRNA genes, Ser2-P and Tyr1-P signal steadily increased along the body of snoRNA genes, with peaks that coincided with the decline in total RNAPII levels downstream of snoRNA poly(A) sites (Fig. 4b). Together, our findings indicate that termination of RNAPII transcription at mRNA and snoRNA genes in *S. pombe* occurs at regions where maximal signal of Ser2 and Tyr1 phosphorylation are detected.

Pcf11 and Seb1 have CTD-interacting domains (CID) that preferentially recognize Ser2-phosphorylated RNAPII in vitro[19,25]. Therefore, we compared the ChIP-seq profiles of Pcf11 and Seb1 to genome-wide RNAPII Ser2-P signal. As shown for a protein-coding and a snoRNA gene (Fig. 4c, d), the peak occupancy of Pcf11 and Seb1 binding generally coincided with regions of maximum Ser2-P signal that were found downstream of poly(A) sites. The average occupancy profiles of Pcf11 and Seb1 also colocalized with regions of maximal Tyr1-P signal (Fig. 4c–f). Globally, ChIP-seq signals of 3′ end processing factors were found to be more correlated genome-wide with Ser2 and Tyr1 phosphorylation than with Ser5 and Ser7 phosphorylation (Figs. 4g, 0.58 vs 0.25). These results support the view that Ser2 and Tyr1 CTD phosphorylation are functionally relevant to fission yeast 3′ end processing and transcription termination.

**ncRNA transcription termination by torpedo in fission yeast.** Cleavage-dependent transcription termination depends on the activity of the conserved XRN2 nuclease (Dhp1 in *S. pombe*). To address whether termination of snoRNA transcription in *S. pombe* was generally dependent on Dhp1, we analyzed RNAPII occupancy in Dhp1-depleted cells. Because deficiencies in Rat1 and XRN2 have been shown to affect RNAPII occupancy[6,37], we introduced a spike-in normalization step[38] using a constant amount of *S. cerevisiae* chromatin to directly compare ChIP-seq samples between wild-type and Dhp1-depleted cells (see Methods for details). The distribution of total RNAPII (Rpb1) displayed increased density at the 3′ end of the *fba1* mRNA and the *snR99* snoRNA in Dhp1-deficient cells (Fig. 5a, b), consistent with read-through transcription. Evidence of delayed transcription termination in the *dhp1* mutant was also noted at snRNA genes, as indicated by a shift of Rpb1 density into the 3′ flanking region of *U5* and *U1* snRNAs (Fig. 5c and Supplementary Fig. 5a). Read-through transcription was also detected by a CTD-independent ChIP approach that used an antibody to the HA-tag of a core

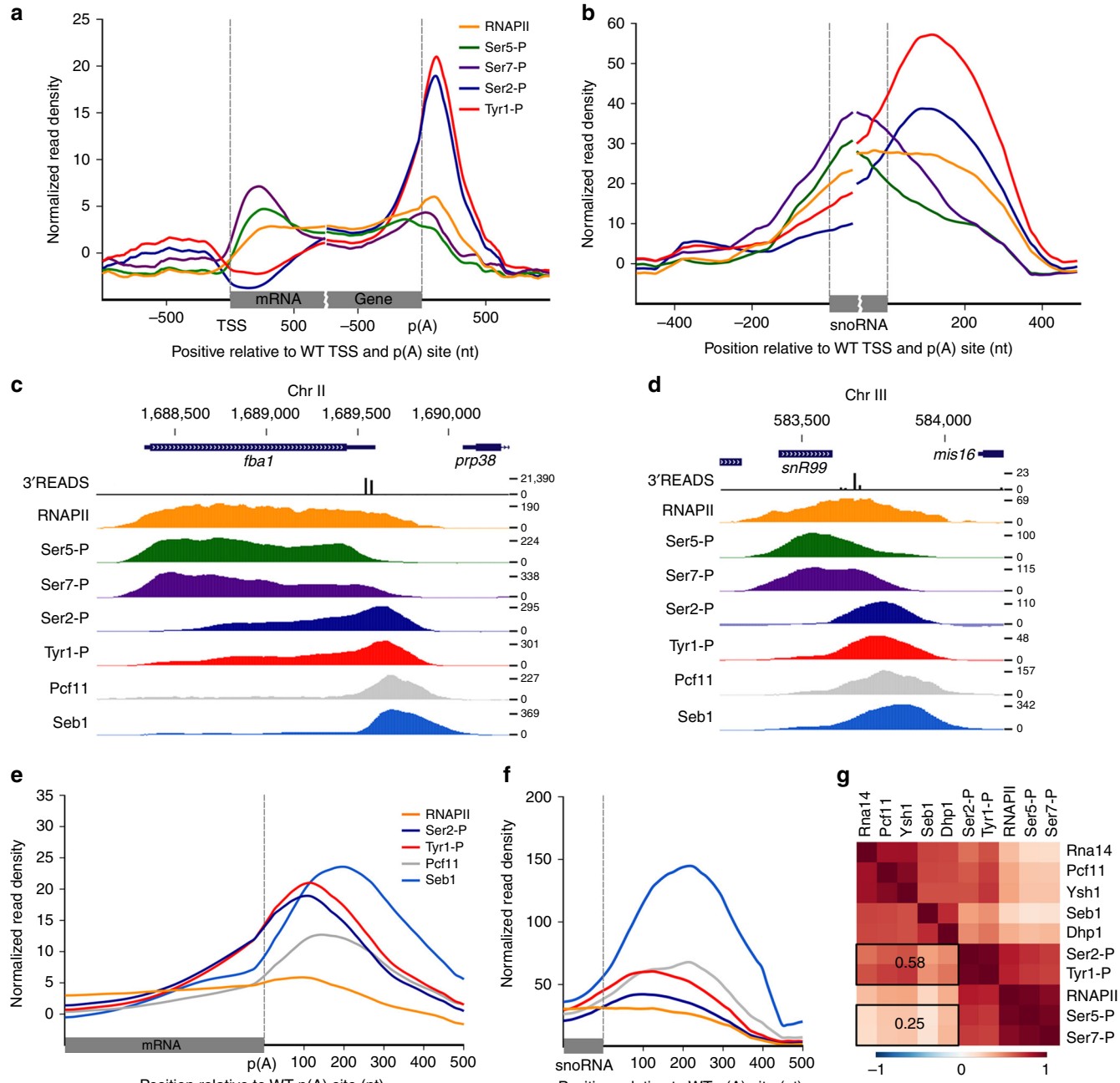

**Fig. 4** Tyr1-P and Ser2-P forms of the RNAPII CTD colocalize with 3′ end processing factors at coding and ncRNA genes. **a, b** Average ChIP-seq profile of total RNAPII (Rpb1) and the indicated CTD modifications in a WT strain across 4755 mRNAs (**a**) and 24 monocistronic snoRNAs (**b**) with a mapped poly (A) site in minimal medium. **c, d** PAS read density (3′READS) and normalized ChIP-seq signal of total RNAPII (Rpb1), the indicated CTD modifications, as well as Pcf11 and Seb1 across the *fba1* mRNA (**c**) and the *snR99* snoRNA (**d**) genes. **e, f** Average profile of total RNAPII (Rpb1), Ser2-P, Tyr1-P, Pcf11, and Seb1 relative to the poly(A) site of 4755 mRNA (**e**) and 24 monocistronic snoRNA (**f**) genes in minimal medium. **g** Genome-wide pairwise Pearson correlation coefficient matrix at a resolution of 10 bp followed by a hierarchical clustering. Rectangles include the average of a subset of correlation values between 3′ processing factors and CTD modifications

RNAPII component (Rpb3-HA; Fig. 5a–c). Importantly, transcription termination defects are a general feature of Dhp1-deficient cells, as seen by a clear shift in the Rpb1 and Rpb3-HA average signals downstream of the noticeable decline observed in wild-type cells (Fig. 5d, e; compare solid and dotted lines). These results show that Dhp1 is required for RNAPII termination at both coding and ncRNA genes.

Hyperphosphorylation of CTD repeats on Ser2 was previously observed in a temperature-sensitive mutant of *S. cerevisiae rat1*[37]. To address how Dhp1 inactivation globally affects CTD phosphorylation at transcribed *S. pombe* genes, we compared Tyr1, Ser2, Ser5, and Ser7 phosphorylation levels between Dhp1-deficient and control cells using spike-in-normalized ChIP-seq. As shown in Fig. 5f, g, inactivation of Dhp1 resulted in a generalized increase in Ser2-P and Tyr1-P signal, which was most noticeable at the 3′ end of genes. Analysis of Ser2-P/total RNAPII ratios at the 3′ end of genes suggests that the levels of Ser2 phosphorylation does not truly increase in Dhp1-deficient cells, but that the CTD remains phosphorylated on Ser2 for an extended period of time after cleavage (see Supplementary

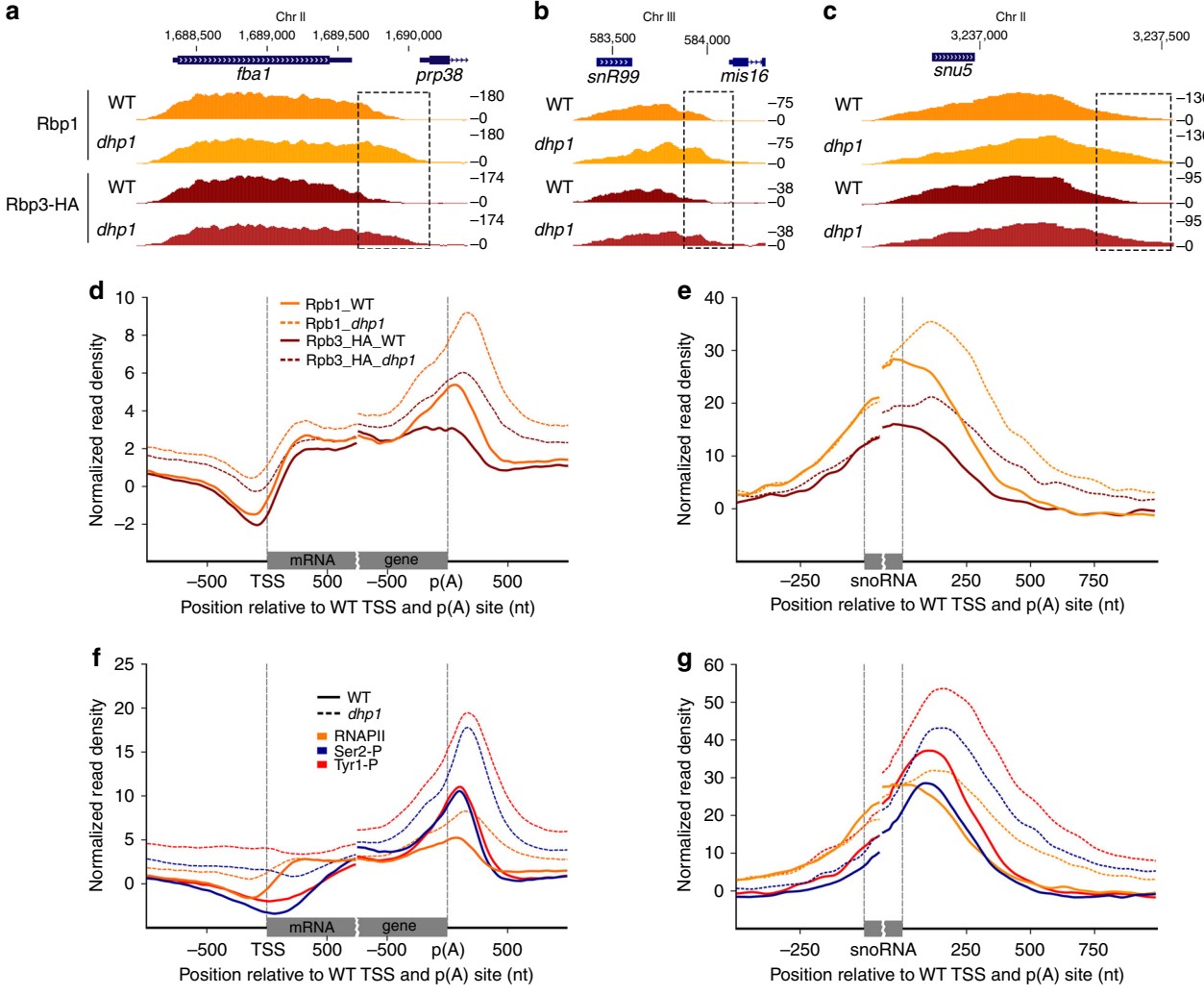

**Fig. 5** The torpedo nuclease Dhp1 is required for transcription termination of coding and ncRNA genes. **a–c** Normalized ChIP-seq signal of RNAPII subunits Rbp1 and Rbp3 in WT and Dhp1-depleted strains across the *fba1* mRNA (**a**), the *snR99* snoRNA (**b**), and the *snu5* snRNA (**c**) genes grown in thiamine-treated minimal medium for 15 h. The dashed-line rectangles highlight delayed transcription termination in Dhp1-deficient cells. **d–g** Average ChIP-seq profile of Rbp1 and Rbp3 (**d**, **e**) or total RNAPII (Rbp1), Ser2-P, and Tyr1-P (**f**, **g**) in WT (solid lines) or Dhp1-depleted (dotted lines) cells across 4755 mRNA (**d–f**) and 24 monocistronic snoRNA (**e**, **g**) genes with a mapped p(A) site in thiamine-treated minimal medium

Fig. 5b). In contrast, Tyr1-P/total RNAPII ratios showed an increase in the maximal signal of Tyr1 phosphorylation in the *dhp1* mutant (Supplementary Fig. 5b), suggesting that more CTD repeats have a phosphorylation mark on Tyr1 in Dhp1-depleted cells compared to wild-type cells. Prolonged Pcf11 binding at the 3′ end of coding (Supplementary Fig. 5c–f) and ncRNA (Supplementary Fig. 5j, k) genes further support the functional relevance of sustained Ser2-P in Dhp1-deficient cells. The changes in CTD phosphorylation at Ser2 and Tyr1 in the *dhp1* mutant do not appear to be the result of reduced recruitment of the Dis2/Glc7 phosphatase, which showed similar binding between wild-type and Dhp1-deficient cells after normalization to total RNAPII (Supplementary Fig. 5g–i). In contrast to Ser2-P and Tyr1-P, Ser5-P and Ser7-P signals were only slightly affected by the depletion of Dhp1 (Supplementary Fig. 5l, m). We conclude that altered CTD phosphorylation is a conserved consequence of Dhp1/Rat1 inactivation.

**snoRNA transcription termination requires Ysh1 cleavage.** The global transcription termination defects at mRNA and snoRNA genes in Dhp1-deficient cells suggest that coding and ncRNA genes are terminated by torpedo in fission yeast. This mechanism entails an endonucleolytic cleavage prior to Dhp1-dependent transcription termination. To test whether the endoribonuclease subunit of the cleavage and polyadenylation specificity factor (CPSF) complex, Ysh1 (CPSF-73 in humans), is required for transcription termination of ncRNA genes in fission yeast, we analyzed RNAPII density in cells where Ysh1 was depleted from the nucleus using an anchor-away strategy. As expected, inactivation of Ysh1 resulted in a clear read-through phenotype at a protein-coding gene (Fig. 6a). Globally, the RNAPII 3′ peak showed a clear shift downstream in the *ysh1* mutant as compared to control cells (Fig. 6d). Importantly, the consequences of Ysh1 nuclear depletion were comparable at the *snR99* gene (Fig. 6b), causing a marked downstream shift in the average RNAPII profile at snoRNA genes (Fig. 6e). *pcf11* and *rna14* mutants also showed clear evidence of read-through transcription at *snR99* and *snR3* genes (Supplementary Fig. 6), consistent with a general role of the CPF in transcription termination at snoRNA genes in fission yeast. The transcription termination defects observed at snoRNA genes in the CPF

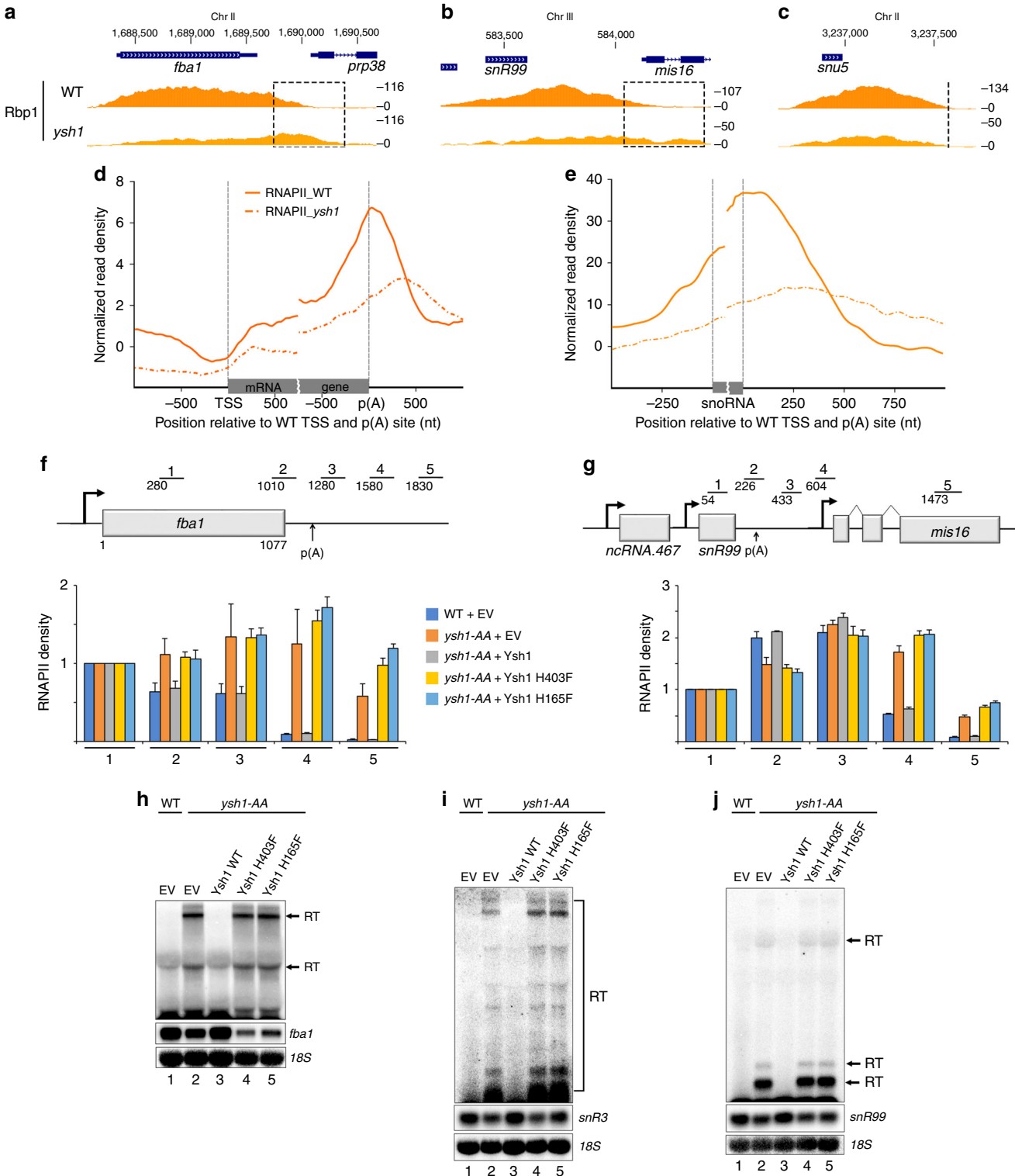

mutants (*ysh1*, *pcf11*, and *rna14*) are thus consistent with the detection of long read-through transcripts (Fig. 2f), which are not detected in the *pab2Δ* mutant (Fig. 2f, lane 8) that does not show transcription termination defects[28]. Interestingly, whereas Dhp1 was required for efficient transcription termination of both snoRNA and snRNA genes (Fig. 5b, c and Supplementary Fig. 5a), the nuclear depletion of Ysh1 did not result in read-through transcription at snRNA genes (Fig. 6c and Supplementary Fig. 7a).

We next addressed whether the endonucleolytic activity of Ysh1 was required to promote termination of snoRNA transcription. We generated *ysh1* alleles expressing single amino acid substitutions at conserved histidine residues that were shown to contact critical zinc atoms in the active site of human CPSF-73[39]. We used a "complementation after nuclear depletion" approach[40] by chromosomal integration of the different *ysh1* alleles into the *ysh1* anchor-away strain and confirmed that the corresponding proteins were expressed (Supplementary Fig. 7b). After nuclear

**Fig. 6** The endonucleolytic activity of Ysh1 is necessary for termination of snoRNA transcription. **a–c** Normalized ChIP-seq signal of total RNAPII (Rbp1) in WT (top) and *ysh1* mutant (bottom) cells across the *fba1* mRNA (**a**), the *snR99* snoRNA (**b**), and the *snu5* snRNA (**c**) genes grown in rapamycin-treated minimal medium for 4 h. The dashed-line rectangles highlight transcriptional read-through at *fba1* and *snR99* genes in Ysh1-deficient cells. **d, e** Average ChIP-seq profile of RNAPII (Rpb1) in WT (solid lines) and Ysh1-deficient (dotted lines) cells across 4755 mRNA (**d**) and 24 monocistronic snoRNA (**e**) genes grown in rapamycin-treated minimal medium for 4 h. **f, g** RNAPII ChIP-qPCR analysis on the *fba1* (**f**) and *snR99* (**g**) genes using extracts prepared from either wild-type (WT) or *ysh1* anchor-away (*ysh1-AA*) strains containing chromosomally integrated constructs that express the indicated versions of FLAG-tagged Ysh1 (WT, H403F, and H165F) as well as an empty vector (EV) control. Bars above the *fba1* and *snR99* genes show the positions of PCR products used for ChIP-qPCR analyses. Cells were grown in the presence of rapamycin for 4 h to deplete endogenous Ysh1 from the nucleus. ChIP signals (percent of input) were normalized to region 1. Error bars indicate SD. *n* = 3 biological replicates from independent cultures. **h–j** Northern blot analysis of *fba1* (**h**), *snR3* (**i**), and *snR99* (**j**) genes using total RNA prepared from using total RNA prepared from the same WT (lane 1) and *ysh1* anchor-away (*ysh1-AA*, lanes 2–5) strains as in **f**, **g**. The position of read-through (RT) transcripts is indicated on the right. The 18S rRNA was used as a loading control

depletion of endogenous Ysh1 by rapamycin, the expression of catalytic mutant versions of Ysh1 resulted in growth arrest (Supplementary Fig. 7c), consistent with the catalytic activity of Ysh1 being required for cell viability[41]. Next, we used RNAPII ChIP-qPCR assays to examine the extent to which the mutant versions of Ysh1 restored the transcription termination defects induced by the nuclear depletion of endogenous Ysh1. As a control, expression of wild-type Ysh1 in the *ysh1* anchor-away strain prevented the rapamycin-dependent increase in RNAPII levels downstream of both protein- and snoRNA-coding genes (Fig. 6f, g, compare dark blue, gray, and orange bars). In contrast, the endonuclease mutant versions of Ysh1 showed increase in RNAPII density at the 3′ end of both mRNA and snoRNA genes (Fig. 6f, g, compare yellow and light blue bars to gray bars). Consistently with these RNAPII ChIP-qPCR data, RNA analysis showed the accumulation of read-through transcripts in cells that expressed mutant versions of Ysh1 (Fig. 6h–j, compare lanes 4–5 to lane 3; Supplementary Fig. 7d). We conclude that the endonucleolytic activity of Ysh1 is required for 3′ end processing and transcription termination of both mRNA and snoRNA genes.

**Transcription termination at long noncoding RNA genes.** Long noncoding RNAs (lncRNAs) contribute a substantial and diverse portion of noncoding transcriptomes. In *S. pombe*, over 85% of the roughly 1500 annotated lncRNAs are expressed below one copy per cell during the mitotic cell cycle[42], thus making it challenging to detect lncRNA expression under normal conditions. Furthermore, a large proportion of the *S. pombe* lncRNAs are antisense to protein-coding genes, rendering analysis of non-stranded ChIP-seq data unreliable. Attempts to obtain a global portrait of CTD phosphorylation patterns and CPF binding at lncRNA genes in fission yeast were unsuccessful, presumably due to their low levels of transcription. We therefore focused our attention on a handful of lncRNA genes that showed robust levels of RNAPII during vegetative growth as determined by our ChIP-seq analysis of Rpb1 and Rpb3-HA, and observed clear enrichment of Ser2-P and Tyr1-P together with binding of CPF and termination factors at the 3′ end of the *SPNCRNA.532* (Fig. 7a) and *SPNCRNA.491* (Fig. 7b) lncRNA genes. Importantly, read-through transcription was detected downstream of *SPNCRNA.532* (Fig. 7a) and *SPNCRNA.491* (Fig. 7b) in cells deficient for Pcf11, Rna14, Ysh1, and Dhp1 (Fig. 7a, b and Supplementary Fig. 6c, d), consistent with a cleavage-dependent mechanism of transcription termination. Similar binding profiles were found for a recently described lncRNA, *SPNCRNA.1459/nam1* (Fig. 7c and Supplementary Fig. 6e), which represses entry into the meiotic differentiation cycle[43]. Although additional work will be required to get a global view of RNA processing and CTD phosphorylation at *S. pombe* lncRNA genes, these data suggest a mechanism of transcription termination that relies on endonucleolytic cleavage and Dhp1-dependent torpedo.

## Discussion

Despite extensive characterization of how the NNS complex promotes termination at ncRNA genes in *S. cerevisiae*, whether NNS-like transcription termination is conserved across eukaryotic species was unclear. Despite the presence of homologs for all of the NNS components, our results indicate that NNS-mediated termination is not conserved in the distantly related yeast *S. pombe*. In the absence of an NNS-like complex, our findings reveal that a universal cleavage-dependent mechanism that involves the conserved Dhp1 5′–3′ torpedo nuclease is used to terminate transcription of both mRNA and ncRNA genes.

Two key indications support the absence of an NNS-like transcription termination pathway in fission yeast. First, Seb1 (Nrd1 homolog), Nab3, Sen1, and Dbl8 do not to form a stable complex in fission yeast[24,25,44]. Second, transcription termination defects were not detected in *nab3Δ*, *sen1Δ*, *dbl8Δ*, and *sen1Δ dbl8Δ* mutants. These data suggest that while NNS components have been conserved between budding and fission yeasts, they have acquired species-specific functions. For instance, *S. pombe* Seb1 recognizes Ser2-phosphorylated RNAPII and is recruited to the 3′ end of both mRNA and ncRNA genes[24,25]. In contrast, *S. cerevisiae* Nrd1 preferentially binds to Ser5-phosphorylated RNAPII and is enriched at noncoding transcription units[2]. Also consistent with the idea that NNS components have functionally diverged between the evolution of budding and fission yeasts from a common ancestor, Sen1 is primarily associated with RNA polymerase III (RNAPIII)-transcribed genes in *S. pombe*[44]. An NNS-like transcription termination pathway is also unlikely to exist in metazoans, as clear homologs or functional analogs of Nrd1 and Nab3 components have not been described. Conversely, a homolog for the yeast Sen1 helicase has been characterized in humans (Senataxin, SETX), and a role for SETX in promoting efficient transcription termination at model protein-coding genes has been described[45,46]. However, how SETX globally contributes to transcription termination of mRNA genes remains to be demonstrated, as data presented here as well as in *S. cerevisiae*[47] indicate that yeast Sen1 is not required for efficient transcription termination at protein-coding genes.

By combining genome-wide studies and functional analyses of the fission yeast mRNA 3′ end processing machinery, we provide evidence supporting that a general cleavage-dependent, torpedo-mediated mechanism is used to promote transcription termination of protein-coding and ncRNA genes. Several observations support this conclusion: (i) mRNA 3′ end processing (Rna14, Pcf11, and Ysh1) and termination (Dhp1) factors are specifically recruited at the 3′ end of independently transcribed snoRNA, snRNA, lncRNA genes; (ii) mRNA 3′ end processing factors are required for snoRNA synthesis; (iii) polyadenylated pre-snoRNAs are produced from most snoRNA genes; (iv) 3′ end *cis*-sequences of protein-coding genes can promote snoRNA 3′ end maturation and accumulation; and (v) transcription termination at ncRNA

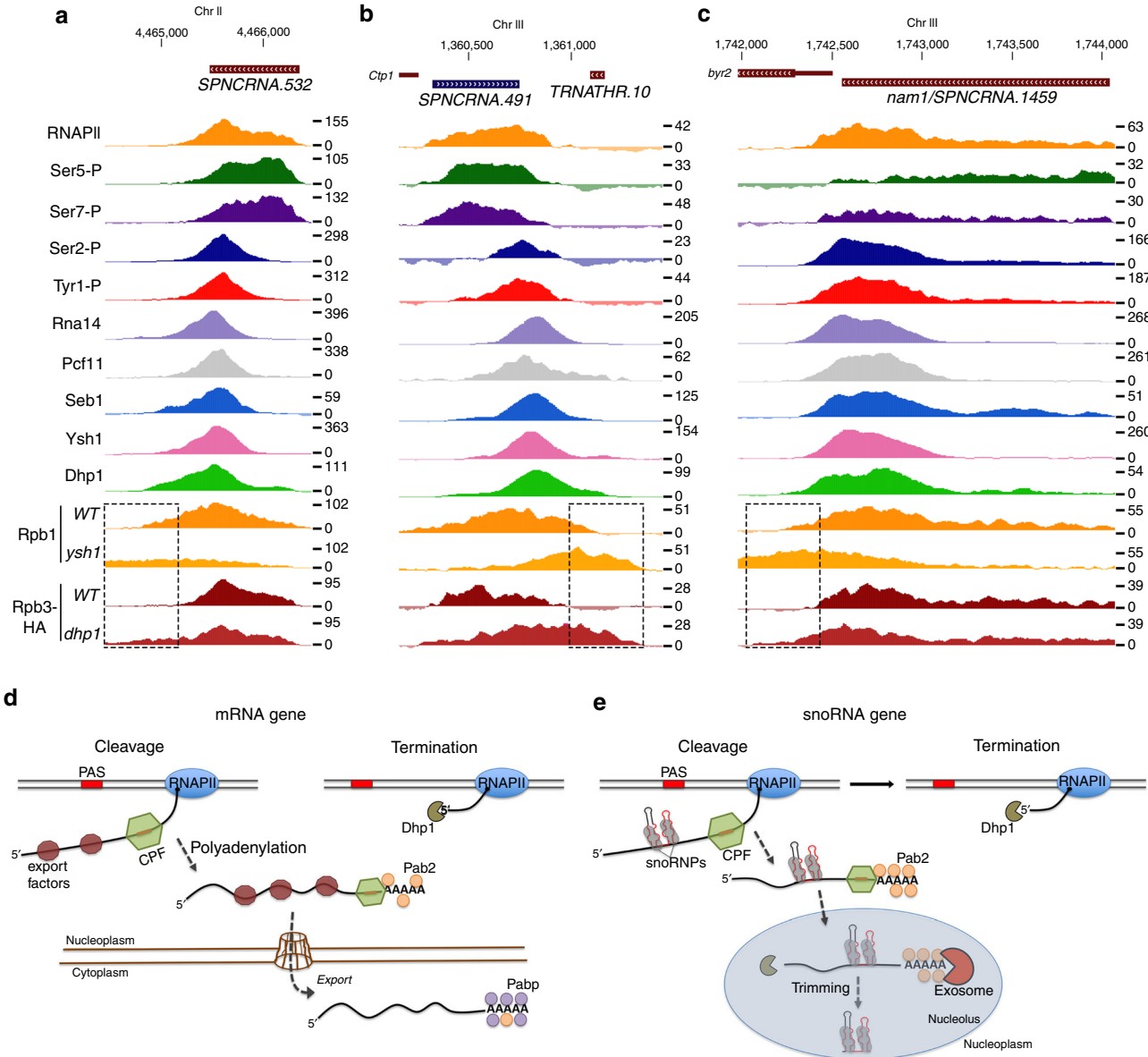

**Fig. 7** Transcription termination at lncRNA genes and general model for 3′ end processing and transcription termination of mRNA and snoRNA genes in fission yeast. **a–c** Normalized ChIP-seq signal of total RNAPII (Rpb1 and Rbp3-HA), the indicated CTD modifications, and 3′ end processing factors across the *SPNCRNA.532* (**a**), the *SPNCRNA.491* (**b**), and the nam1/*SPNCRNA.1459* (**c**) genes. **d**, **e** Recruitment of cleavage and polyadenylation factors (CPF) by poly(A) signals (PAS) is a common feature of mRNA and snoRNA genes (see *Cleavage*). Endonucleolytic cleavage by the CPF-associated Ysh1 nuclease will generate an RNAPII-bound unprotected 5′ end that will be targeted by the 5′–3′ exonuclease Dhp1, contributing to termination of mRNA and snoRNA transcription (see *Termination*). The RNA-bound CPF complex promotes snoRNA 3′ end maturation, possibly by facilitating Pab2 recruitment to snoRNA precursors. The co-transcriptional recruitment of specific export factors to nascent mRNAs (**d**) may represent a decisive step that prevents Pab2-dependent exosome-mediated RNA processing in the nucleus (**e**)

genes requires endonucleolytic cleavage and the torpedo nuclease Dhp1. These data support a model in which cleavage and polyadenylation are essential steps for 3′ end processing and transcription termination at both mRNA and snoRNA genes. Although it has been shown that particular components of the mRNA 3′ end processing machinery, such as Pcf11, Rat1, and Glc7, are recruited to short ncRNA genes in *S. cerevisiae*[16,48–50], these factors appear to function either separately from the CPF complex to contribute to NNS-dependent termination or as a failsafe mechanism of transcription termination. This contrasts our findings in fission yeast, where the prevailing mechanism of transcription termination at ncRNA genes involves the cleavage and polyadenylation complex. Given the unexpected similarity of cleavage-dependent transcription termination at mRNA and snoRNA genes, it will be interesting to understand the mechanism used to distinguish between mRNAs and snoR-NAs; whereas the mature mRNA needs to retain its poly(A) tail, the polyadenylated 3′-extended pre-snoRNA is targeted by Pab2-dependent exosome-mediated 3′ end trimming to yield a mature non-polyadenylated snoRNA. We speculate that co-transcriptional decoration of nascent pre-mRNAs with nuclear export factors prevents nuclear retention and Pab2-dependent 3′ end trimming, whereas the nucleolar-targeting signals of snoRNA ribonucleoproteins (snoRNPs) assembled during snoRNA transcription[51] would promote nuclear retention and 3′ end maturation.

In addition to the detection of read-through transcription at snoRNA genes, depletion of CPF components including Pcf11, Ysh1, and Rna14 also led to the accumulation of polyadenylated snoRNA precursors similar to the *pab2Δ* mutant. Thus far, our studies on 3′ end processing and transcription termination at snoRNA genes have identified three types of mutants: (i) one type showing transcription termination defects but no detectable RNA-processing deficiencies (*dhp1*), (ii) one type showing RNA maturation defects but normal transcription termination[28] (*pab2*), and (iii) mutants that show defects in both transcription termination and snoRNA 3′ end processing (*pcf11, ysh1, rna14,* and *seb1*). Collectively, these findings suggest that in addition to promoting the endonucleolytic cleavage for Dhp1-dependent termination, the CPF also contributes to snoRNA 3′ end processing in fission yeast. Although the precise role of the CPF in snoRNA 3′ end maturation remains to be determined, it may be important for the proper recruitment of Pab2, a view that is supported by protein interaction studies[52,53].

As for snoRNA genes, termination of snRNA transcription in *S. cerevisiae* appears primarily dependent on NNS[54]. We found that whereas transcription termination at both snoRNA and snRNA genes required Dhp1 in *S. pombe*, termination of snRNA transcription was not affected in Ysh1-deficient cells. This result suggests that the 5′ entry point for Dhp1 loading at snRNA genes may be mediated independently of the nuclease activity of the mRNA 3′ end processing machinery. Interestingly, read-through transcription at the *S. pombe* U2 snRNA gene was previously reported in a temperature-sensitive mutant of *pac1*[55], which encodes a homolog of the *S. cerevisiae* RNase III-like endonuclease Rnt1. Although the identity of the nuclease responsible for cleavage at *S. pombe* snRNA genes remains to be determined, our data suggest that snRNA transcription termination in fission yeast is more closely related to humans than to *S. cerevisiae*, as mRNA 3′ end processing factors[56] and the torpedo nuclease XRN2[6] are important for efficient transcription termination of human snRNA genes.

Our study also elicits interesting questions as to the functional significance of RNAPII CTD phosphorylation in fission yeast. First, given the reciprocal relationship between Ser2 phosphorylation and 3′ end processing in *S. cerevisiae* and humans[14], the peak of Ser2-P signal at the 3′ end of both mRNA and ncRNA genes provides an additional line of evidence supporting co-transcriptional 3′ end processing by the CPF in fission yeast. Intriguingly, however, Ser2-P does not appear to be essential in *S. pombe*[57,58], suggesting that protein-RNA interactions may function as a redundant mechanism of CPF recruitment, as recently shown for elongation factors in budding yeast[59]. Second, our data indicates that the pattern of Tyr1-P on the CTD of *S. pombe* RNAPII at the 3′ end of genes differs from the distribution of Tyr1-P in *S. cerevisiae*[20]. Accordingly, although Tyr1-P levels increase along coding regions in both budding and fission yeasts, Tyr1-P signal decreases upstream of the poly(A) site in *S. cerevisiae*, whereas Tyr1-P signal persists downstream of the poly(A) site in *S. pombe*. These differing patterns of Tyr1-P around the poly(A) were unexpected, as Tyr1-P in *S. cerevisiae* was previously proposed to inhibit binding of CID-containing termination factors (Pcf11 and Rtt103) to Ser2-P CTD and prevent premature termination in coding regions[20], a role that would also be anticipated to contribute to general transcription in *S. pombe*. Yet, the similar distribution of Ser2-P and Tyr1-P profiles downstream of the poly(A) site in *S. pombe* argues for diverging roles of Tyr1-P between budding and fission yeasts. Consistent with this idea, a truncated version of the CTD in which a phenylalanine replaces Tyr1 (Y1F) in each heptad repeat results in lethality in *S. cerevisiae*[60], whereas the analogous truncated Y1F CTD mutant is viable in *S. pombe*[58].

Our genome-wide analysis of RNAPII occupancy revealed that Dhp1 is necessary for transcription termination of most protein-coding genes in fission yeast. In addition to transcription termination defects, elevated levels of Ser2-P signal were found at model protein-coding genes in a temperature-sensitive mutant of *S. cerevisiae rat1*[37]. Our data extend these finding by showing that (i) this observation is conserved in fission yeast, (ii) that Dhp1 depletion also causes elevated levels of Tyr1 phosphorylation, and (iii) that Dhp1 deficiency affects Tyr1 and Ser2 CTD phosphorylation at the genome-wide level. Although further studies will be needed to investigate how Dhp1/Rat1 nucleases modulate CTD phosphorylation at the 3′ end of genes, which may involve competition with the recruitment of CTD kinases[61], this finding highlights the important functional relationship between CTD phosphorylation and transcription termination.

Given that NNS-like transcription termination does not appear to be conserved in mammalian cells, it remains unclear how independently transcribed human snoRNAs, such as U3, U8, U13, and the human telomerase RNA, acquire their mature 3′ end. The fact that we have recently uncovered a polyadenylation-dependent 3′ end maturation pathway for the human telomerase RNA that depends on the nuclear poly(A)-binding protein PABPN1[62] supports the existence of an evolutionarily conserved role for Pab2/PABPN1 in the maturation of independently transcribed snoRNAs. It will be interesting to determine whether 3′ end processing and transcription termination of human snoRNA genes that are characterized by independent transcription unit depend on the mRNA cleavage and polyadenylation complex, as unveiled by our findings using fission yeast.

## Methods

**Yeast strains and media.** A list of all *S. pombe* strains used in this study is provided in Supplementary Table 1. Cells were routinely grown at 30 °C in Edinburgh minimal media (EMM) or in yeast extract medium (YES) supplemented with adenine, uracil, histidine, and leucine. Conditional strains, in which the genomic copy of the essential genes *pcf11, dhp1,* and *seb1* are expressed from the thiamine-sensitive *nmt41* and *nmt81* promoters (Supplementary Table 1), were repressed by the addition of 60 μM of thiamine in the EMM media for 12–15 h as previously described[24,29]. Depletion of nuclear Ysh1 and Rna14 were done by an anchor-away strateby[32]. Briefly, wild-type (WT), Ysh1-FRB-GFP (*ysh1-AA*), and Rna14-FRB-GFP (*rna14-AA*) strains were grown in either EMM without leucine or YES media, and rapamycin was added or not at a final concentration of 2.5 μg/ml for 4–15 h. Cells were collected at OD$_{600\mathrm{nm}}$ of ~0.5–0.8. Gene disruptions and C-terminal tagging of proteins were performed by PCR-mediated gene targeting[63] using lithium acetate method for yeast transformation. Expression of tagged proteins was confirmed by western blotting and knockouts strains by the absence of RNA by RT-PCR.

**Growth assays.** Exponential cultures of *S. pombe* cells were adjusted to an OD$_{600\mathrm{nm}}$ of 1.0, and serially diluted tenfold using water. Each dilution was spotted (3 μl per spot) on EMM plates with or without 15 μM of thiamine in the case of strains using the thiamine-sensitive *nmt* promoter ($P_{nmt}$), or on YES plates containing DMSO or 2.5 μg/ml rapamycin for the analysis of anchor-away mutants. Plates were incubated at 30 °C for 2–7 days.

**Microscopy.** Ysh1-GFP and Rna14-GFP localization was detected by using fluorescence microscopy. Briefly, precultures grown in EMM supplemented with adenine, uracil and histidine (EMM Leu⁻) were used to inoculate larger 25-ml cultures that we grown in EMM leu⁻ to early log phase (OD$_{600\mathrm{nm}}$ 0.25–0.3). Rapamycin was then added to a final concentration of 2.5 μg/ml (same volume of DMSO was added in parallel as control) and samples were taken at 0, 0.5, 1, 2, 3, 4, and 5 h, which were diluted 1:10 in water. Nuclei were stained using Hoechst 33342 for 5 min (0.2 mg/ml) and live cells were mounted on a 1.2% agarose/EMM leu⁻ pad as described previously[64]. GFP-tagged proteins and nuclei were detected at 470 nm and 365 nm, respectively, using a Colibri system (Carl Zeiss Canada, Toronto, ON, Canada) on a Zeiss Axio Observer Z1 inverted microscope with a ×60/1.4 oil objective. Data were analyzed using the ZEN black software (Carl Zeiss Canada).

**RNA preparation and analyses.** Total RNA was extracted using the hot-acid phenol method. RNA samples were resolved on agarose-formaldehyde gel or on 6% polyacrylamide-8M urea gels, transferred onto nylon membranes and probed as

described previously[24]. RNase H assays were performed as described previously with 15 μg of RNA[28]. Briefly, total RNA was RNAse H-treated in a mixture containing a snoRNA-specific complementary oligonucleotide to induce cleavage and release a specific 3′ fragment, as well as with or without oligo d(T) to remove the heterogeneity of poly(A) tails. Preparation of probes, blot hybridization, washes, and quantification of signals were done as described previously[24]. Uncropped blots are shown in Supplementary Fig. 8.

**Chromatin immunoprecipitation (ChIP) assays.** ChIP-qPCR and ChIP-seq experiments were performed as described previously[24]. Chromatin was immunoprecipitated directly using Pan Mouse IgG Dynabeads (Life Technologies, 11041) for TAP-tagged proteins. ChIP were also performed using Pan Mouse IgG Dynabeads coated with antibodies specific to Rpb1 (clone 8WG16; Covance, MMS-126R) or the hemagglutinin (HA) sequence (clone 12CA5, Roche, 11 583 816 001). ChIP with phospho-specific CTD antibodies were performed using 2 μg of antibody: Ser2-P (Abcam, ab5095) and Ser5-P (Abcam, ab5131) were coated to Dynabeads M-280 Sheep anti-rabbit IgG beads (Life Technologies, 11203D), whereas Tyr1-P (clone 3D12, Active Motif, 61383) and Ser7-P (clone 4E12, EMD Millipore, 04–1570) were coated to Dynabeads Protein G beads (Life Technologies, 10003D). Control ChIP assays with untagged strains or with isotype matched control antibody were performed throughout the study.

ChIP-seq profiles of total RNAPII (Rpb1; clone 8WG16) in WT, nab3Δ, sen1Δ, dbl8Δ, and sen1Δ/dbl8Δ strains were performed in YES media as described previously[24]. ChIP-seq experiments of total RNAPII and phospho-CTD modifications in wild-type and dhp1-depleted cells were performed with a spike-in adjustment procedure that allows a quantitative comparison between independent strains or conditions[38]. Chromatin preparations of S. pombe FBY1995 (Rpb3-HA; WT) and FBY2016 (Rpb3-HA; P_nmt-dhp1) strains and S. cerevisiae FBY2064 strain (reference strain containing a copy of HA-tagged histone H2B) were done as a regular ChIP, except that chromatin extracts have been frozen in liquid nitrogen and stored at −80 °C before affinity purifications. For Tyr1-P, Ser2-P, Ser5-P, Ser7-P, and 8WG16 IPs, a S. pombe/S. cerevisiae chromatin ratio of 0.9:0.1 was used, whereas a chromatin ratio of 0.995:0.005 was used for Rpb3-HA purifications. Following combination of S. pombe/S. cerevisiae chromatin extracts, ChIP assays were performed as described above using antibody-coated beads. ChIP-seq profiles of total RNAPII from WT and ysh1-AA strains were also performed with a spike-in adjustment (ratio S. pombe/S. cerevisiae of 0.9:0.1) using extracts from S. pombe that were previously grown in EMM and treated for 4 h with 2.5 μg/ml of rapamycin. Similarly, ChIP-qPCR of Rpb1 in ysh1 nuclease mutants (Fig. 6) were grown in EMM and treated with rapamycin for 4 h.

**Protein analysis.** Protein analysis was essentially performed as described[24]. Briefly, total extracts were prepared by lysis of mid-log phase S. pombe cultures in ice-cold lysis buffer (50 mM Tris-HCl (pH 7.5), 5 mM MgCl₂, 150 mM NaCl and 0.1% NP-40 supplemented with 1 mM PMSF and 1× PLAAC) using a Precellys 24 homogenizer system (Bertin Technologies). Clarified lysates were normalized for total protein concentration using Bradford protein assay. Total extract proteins were separated by SDS-PAGE, transferred to nitrocellulose membranes, and analyzed by immunoblotting using antibodies against the hemagglutinin (HA) protein (Roche, 11 583 816 001; 1:500 (v/v) dilution), the GFP protein (11 814 460 001; 1:500 (v/v), the protein A tag (Sigma-Aldrich, P3775; 1:10,000 (v/v) dilution), the FLAG peptide (Sigma-Aldrich, F1804; 1:500 (v/v) dilution) and the α-tubulin protein (Sigma-Aldrich, T5168; 1:1000 (v/v) dilution). Membranes were then probed with goat anti-rabbit or anti-mouse secondary antibodies conjugated to IRDye 800CW (LI-COR, 926–32213; 1:15000 (v/v) dilution) and AlexaFluor 680 (Life Technologies, A-21057; 1:15000 (v/v) dilution), respectively. Protein detection was performed using an Odyssey infrared imaging system (LI-COR).

**snR99 constructs with different terminators.** A list of all plasmids and primers used in this study is provided in Supplementary Table 2 and Supplementary Table 3, respectively. snR99 constructs containing snR99, trx1, or rps2 terminator sequences were generated using the ade6 integration vector (pFB366) containing 1-kb of snR99-specific 5′ promoter sequences followed by the snR99 snoRNA region. The snR99 snoRNA region was then fused to either 1-kb of snR99 3′ sequences (pFB600), of trx1 3′ sequences (pFB622), or rps2 3′ sequences (pFB652). All constructs were confirmed by DNA sequencing. For single integration into the ade6 locus, pFB600, pFB622, and pFB652 were linearized by digestion with the AatII restriction enzyme and transformed into a snR99-null strain. Positive integrants were confirmed by growth selection on EMM agar plates lacking adenine and expression were confirmed by RT-qPCR using primers located in the snR99 region. Deletion of pab2 was than generated in these strains using the kanMX6 marker.

**Ysh1 expression constructs.** The wild-type Ysh1 expression constructs were created by a three-step cloning procedure using the ade6 integration plasmid (pFB366) as a host vector containing 640-bp of ysh1 5′ UTR sequences, the ysh1 coding sequence including introns, and 846-bp ysh1 3′ UTR sequences. Fusion of a 3x FLAG tag to C terminus of Ysh1 was performed using Q5 site-directed mutagenesis using primers containing the 3× FLAG tag DNA sequence, generating

plasmid pFB1337. To generate amino acid substitutions at residues involved in Ysh1 endonuclease activity, histidine (H) residues 165 or 403 of Ysh1 were substituted to phenylalanine (F) to create pFB1355 (H165F) and pFB1358 (H403F), respectively. All mutations were performed by site-directed mutagenesis using pFB1337 as a template. All DNA constructs were confirmed by DNA sequencing. For single integration into the ade6 locus, pFB366, pFB1337, pFB1355, and pFB1358 were linearized and transformed into FBY2066 and/or FBY2110 strains. Positive integrates were confirmed by growth selection on EMM agar plates lacking adenine and leucine and by western blotting.

**Library preparation and Illumina sequencing.** DNA libraries for ChIP-seq experiments were prepared as described previously[24] using either the NEBNext ChIP-Seq Library Prep Master Mix Set for Illumina kit (New England BioLabs) or the SPARK DNA Sample Prep Kit Illumina Platform (Enzymatics) according to the manufacturer's instructions.

**ChIP-seq processing.** Briefly, the raw reads were trimmed using Trimmomatic version 0.32[65] with param ILLUMINACLIP:2:30:15 LEADING:30 TRAILING:30 MINLEN:23, and quality inspection was conducted using FastQC version 0.11.4 (https://www.bioinformatics.babraham.ac.uk/projects/fastqc/). Considering that most of the samples were generated from chromatin contain exogenous spiked-in from S. cerevisiae, the trimmed reads from all data sets were aligned using BWA version 0.7.12-r1039[66] with the algorithm mem or aln depending on the read length (above or below 70 nt) and default parameters, onto a concatenated reference genome containing both the sequences of the S. pombe ASM294v2 and S. cerevisiae sacCer3 assemblies, inspired by others[38]. Note that no filtering on mapQ was performed in order to avoid discarding the signal at regions of the genome that are duplicated (BWA is randomly assigning the reads), but we generated mappability tracks for various read length to help the interpretation of particular regions. The number of reads mapped on the concatenated genomes varied from 91.5 to 99.6% of the sequenced reads (Supplementary Data 3). Note that <0.28% of reads coming from samples without spike-in were aligning to the S. cerevisiae genome (Supplementary Data 3). Signal density files in BedGraph format were then generated using BEDTools genomecov version 2.17[67] with default parameters, then converted in uniform 10 nt bins WIG files for further normalization steps (inspired by the script bedgraph_to_wig.py [https://gist.github.com/svigneau/8846527]).

**ChIP-seq normalization.** Each signal density file of the data sets without spike-in was scaled such that the total sum of the signal over the S. pombe genome is equivalent to 1 M reads of 100 nt, then the signal of the input data set was subtracted from its corresponding IP data set to generate the "sclWT-ctrl" files. The normalization of the data sets with spike-in was inpired by the method of presented by Orlando et al.[38]. For each data set, the sum of the signal coming from the reads aligned to the S. cerevisiae genome was first scaled to the equivalent of 1 M mapped reads of 100 nt, and the same scaling factor was applied to the signal coming from the reads aligned to the S. pombe genome to generate the "normSI" files. For the data sets generated in a WT and mutant strains, the S. pombe signal was next scaled to a total signal equivalent to 1 M mapped reads of 100 nt and the same scaling factor was applied to the corresponding mutant data set to generate the "norm-SI_sclWT" files. The signal of the input data set was then subtracted from its corresponding IP to generate the final "normSI_sclWT-ctrl" files used in downstream analyses.

The normalized WIG files were then encoded in bigWig format using the Kent utilities[68] Visual inspection of the data were performed using an AssemblyHub on the UCSC Genome Browser[69] (https://genome.ucsc.edu/cgi-bin/hgHubConnect?hubUrl=https://datahub-i8kms5wt.udes.genap.ca/CTD_sno_pombe/hub&hgHub_do_firstDb=on&hgHub_do_redirect=on&hgHubConnect.remakeTrackHub=on).

**3′READS analysis.** As described in Liu et al.[34], 3′READS reads were mapped onto S. pombe ASM294v2 assembly using Bowtie 2 (local mode)[70]. Uniquely mapped reads (with MAPQ score >10) that had at least two additional 5′ Ts after genome alignment were considered as polyA sites (PAS) reads. Each PAS was then assigned to the closest gene, including ncRNAs. The GTF genome annotations file version ASM294v2.29 was downloaded from PomBase[71,72] in May 2017. However, based on manual inspection of our ChIP data, the orientation of three snoRNAs (namely, SPSNORNA.36, SPSNORNA.37, and SPSNORNA.41) was inverted. In more details and based on Liu et al.[34], PAS supported with at least two reads outside CDS were associated to the closest 3′ end gene annotation in a strand-specific manner within 1 kb, whereas PAS inside CDS were ignored (except for overlapping genes if the closest 3′ end was within 500 pb). For each gene, the strongest PAS identified in the WT strain grown in minimal medium was used to modify the transcriptional 3′ end gene coordinate provided for mRNA genes (Supplementary Data 1) and snoRNA genes (Supplementary Data 2). For each sample, the PAS read density was normalized by reads per million (RPM) prior to be used in Fig. 3.

**ChIP-seq average profiles.** The Versatile Aggregate Profiler (VAP) tool[73,74] version 1.1.0 was used to generate the average profiles with the following common parameters: annotation mode, absolute analysis method, 10 bp windows size, mean

aggregate value, smoothing of 6, and missing data were considered as "0" (with the exception of Fig. 1a where windows size was set to 50 pb). We used four reference points for most of the figures to avoid contamination from adjacent genes; in such a case there are five blocks of data corresponding respectively to the upstream gene (signal ignore), the upstream intergenic region (signal aligned toward the TSS of the gene of interest), the gene of interest (signal split such that the first half was aligned toward the TSS and the second half toward the 3′end/polyA), the downstream intergenic region (signal aligned toward the 3′end/polyA), and the downstream gene (signal ignore). For figures showing either the average profile of data sets over the complete 4755 mRNA genes with associated PAS (Figs. 1a and 4a), the 31 monocistronic snoRNA genes (Fig. 1b) or the 24 monocistronic snoRNA genes with identified PAS (Fig. 4b), only signal in the first and last blocks were ignored. Similarly, for figures focusing only on the 3′ region of the 4755 mRNA genes with associated PAS (Figs. 1f and 4e), the 31 monocistronic snoRNA genes (Fig. 1g) or the 24 monocistronic snoRNA genes with identified PAS (Fig. 4f), only the signal in the second half of the third block (corresponding to the gene of interest) and the fourth block (downstream intergenic region) was shown. To better represent the extension of signal in the mutant strains (Figs. 5d–g, 6d, e, and Supplementary Fig. 5a,l-m), we used only two reference points where the first block contains the signal in the upstream intergenic region potentially contaminated with signal from the upstream gene (signal aligned as described previously), the second block contains the signal of the gene of interest, and the third block the downstream intergenic region also potentially contaminated with signal from the downstream gene. Genome-wide Pearson correlation coefficients (Fig. 4g and Supplementary Fig. 4) were calculated using the epiGeEC tool version 1.0[75].

**Code availability**. All scripts used for data processing and statistical analysis were written in Python, Perl, or R, and are available upon request.

## Data availability

Sequencing data sets have been deposited in the GEO database under accession numbers GSE115595 and GSE95139 for ChIP-seq and 3′READS, respectively. All other data generated or analyzed during this study are included in this published article and its Supplementary Information or from the authors on reasonable request.

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

## Acknowledgements

We thank Michelle Scott and François Robert for critical reading of the manuscript; Luc Gaudreau for *S. cerevisiae* strains; the sequencing platforms of the McGill University and Génome Québec Innovation Centre; Calcul Québec and Compute Canada provided the computing infrastructure used to analyze the data. This work was supported by funding from the Natural Sciences and Engineering Research Council of Canada (NSERC) to F.B. (RGPIN-2017–05482) and P.-É. J. (435710–2013), and by funding from the National Institute of General Medical Sciences to B.T. (GM084089); P.-É.J. is supported by the Fonds de recherche du Québec – Santé (FRQS) and F.B. holds a Canada Research Chair in Quality Control of Gene Expression.

## Author contributions

M.L. and F.B. conceived the study and the experimental frame, while M.-A.R. and P.-E.J. planned and designed the pipelines for processing the genome-wide data. M.L. prepared chromatin extracts for ChIPs, performed ChIP-seq, including QCs and library pre-parations. M.L. performed most of the RNA analyses with help from J.-N.H, J.-N.H. made the Ysh1 constructs and performed the phenotypic characterization of the *ysh1* and *rna14* anchor-away mutants. X.L. prepared the 3′READS libraries with help from B.T., D.M. and S.R. helped with the preparation of the ChIP-seq libraries. M.-A.R. and P.-E.J. performed all of the bioinformatics analyses of the ChIP-seq and 3′READS data with the help of X.L., M.L., M.-A.R., P.-E.J. and F.B. prepared and finalized the figures. F.B. wrote the manuscript with help of P.-E.J., which was reviewed by all authors.

## Additional information

**Competing interests:** The authors declare no competing interests.

