## [Peer Review File · Nature Communications]

Reviewers' comments:

Reviewer #1 (Remarks to the Author):

In this article Larochelle et al. address the mechanism of transcription termination for snoRNAs in *S. pombe*. It is well established that in *S. cerevisiae* transcription termination for snoRNAs (and some snRNAs) depends on the Nrd1-Nab3-Sen1 (NNS) complex. The authors have previously shown that the *S. pombe* homologue of Nrd1, Seb1, is required for efficient transcription termination at mRNA coding genes, and that Nab3 and Sen1 are not involved in termination at a model snoRNA gene. The authors extended this negative finding to a genomewide perspective, by analyzing the Pol II ChIP seq signal in *sen1Δ* and *nab3Δ* cells. A transcription termination defect was found at snoRNAs upon depletion of Seb1, suggesting that the mRNA termination pathway could be required at these genes. In line with this hypothesis they convincingly show that Rna14, Pcf11, Ysh1 and Dhp1, all involved in mRNA termination, localize also at the end of independently transcribed snoRNA genes, and that the presence of these factors is required for the normal production of a few snoRNAs. Notably, they detected decreased level of mature snoRNA, increased precursors and longer RNA species upon depletion of Rna14, Pcf11, and Ysh1. The occurrence of transcription termination defects at snoRNAs is supported by more direct analyses of Pol II distribution upon depletion of Ysh1 and Dhp1. The authors conclude that the function of the *S. cerevisiae* NNS complex is not conserved in *S. pombe* and that in this organism termination at snoRNAs depends on the pathway terminating transcription at protein coding genes.

This is an important and novel conclusion and could justify further consideration of this article in Nature Communication. However, a few major concerns should be addressed before.

Major points

In figure 2 the authors show that depletion of Pcf11, Ysh1 and Rna14 leads to the detection of polyadenylated snoRNA precursors, very similar to the ones detected in *pab2Δ* cells. The increased detection of precursors is not expected in termination mutants. This is seemingly at odd with a simple model according to which these factors cleave the nascent RNA to generate the precursors and provide an entry point for Dhh1. It is unclear how cleavage would be generated in the absence of Ysh1.

If the termination defect is due to a cleavage defect (as suggested by the implication of Ysh1) the production of snoRNA precursors is expected to decrease to the benefit of longer species, with presumably poorly defined 3' ends. But the opposite is apparently observed

The 3'-extended species shown in figure 2 could be primary termination products and generated independently from CPF-dependent cleavage. Still, why are they only visible when the CPF is depleted? Is the CPF involved in the maturation of snoRNA precursors?

Similarly, why the first PAS signal of snR99 – which is supposed to be the PAS used in wild type cells to induce termination – is strongly selected in Pcf11-depleted cells (Fig3B)? Or is it always selected with the same efficiency and the RNA not processed in the absence of Pcf11?

The issue of processing versus termination could be addressed by coupling a processing defect (maybe *pab2Δ*) for visualizing the precursors to the putative termination defect and showing that the precursors disappear when termination is affected.

It is also unclear what is the origin of the long RT species shown in figure 2F. These could be evidence of readthrough transcription – as the authors suggest – but also additional processing sites that are only detected in mutants (akin to the 3'-extended species of figure 2B-E). It is essential that the authors clarify these points and provide a more direct evidence of transcription termination defects

(i.e. polymerase detection) in Pcf11 and Rna14 cells.

The authors use "ncRNAs" throughout the paper while in reality they only show data concerning snoRNAs (and one snRNA). But many additional non-coding RNAs are also present in *S. pombe*: is transcription termination also affected at other classes of non-coding RNAs in CPF mutants? They should perform such an analysis before concluding that the function of the *S. cerevisiae* NNS complex is not conserved in *S. pombe* and that the CPF terminates transcription of ncRNA genes. This might also help clarifying the question of processing vs termination raised above.

Reviewer #2 (Remarks to the Author):

The paper by Larochelle et al makes a convincing case for a novel form of 3' end formation and termination of transcription at the 3' ends of non-coding RNAs in fission yeast. The data in support of the conclusions are thorough and of high quality. The conclusions for the most part are compelling and important because they establish a significant diversity in the mechanisms responsible for ncRNA 3' end formation with substantial differences between budding and fission yeast. I have only minor suggestions.

1. It would be good to provide some explanation for the difference in termination sites between the two WT strains in Figs 1A, B.
2. It is counter-intuitive that mutation of the cleavage-polyadenylation factors Pcf11, Ysh1, and Rna14 should lead to accumulation of polyadenylated snoRNA transcripts and the authors should provide an interpretation of this result to help the reader.
3. In figure 3 do the cleavage sites in the chimeric snR99-rps2 and snR99-trx1 genes correspond to those used in the natural Rps2 and Trx1 mRNAs?
4. The use of the term "rapid bursts of Ser5-P and Ser7-P" line 32 p. 9 is inappropriate. This is not a kinetic experiment.
5. The authors should acknowledge the reported localization of Rat1 on snoRNA genes in budding yeast (ref. 18) in common with the results reported here for fission yeast.

Reviewer #3 (Remarks to the Author):

In this manuscript, the Bachand laboratory builds on his previous description of Seb1/Nrd1. They report that contrary to budding yeast, a common mode of transcription termination relying on cleavage produces distinct types of polyadenylated and non-polyadenylated RNAs in fission yeast. The paper is well-written and mainly descriptive and several conclusions rely on pure correlation only. The absence of a NNR complex in fission yeast is expected based on the work of this lab (G&D, 2016) and the Vasilieva lab (Nat Com, 2017). Previous work from the Murphy lab has also reported the use of polyadenylation factors at snRNA genes in human (NAR, 2014). Therefore, it is not completely clear that the work presents enough novelty to justify publication in a non-specialized journal. In addition, major technical issues impede publication of the current manuscript.

minor comments

- the use of the word « non-coding RNA (ncRNA) » while referring to snoRNAs and snRNAs is confusing. There are many more ncRNAs (about 1500) in fission yeast that do not belong to this category and whose transcription is similar to the coding RNAs. In addition, rRNAs and tRNAs are also obviously ncRNAs.
- some metagene analyses are difficult to visualize due to the number of tracks and use of similar colours to distinguish them.
- in Figure 1, it would make sense to show the peak of S2P normalized on PolII (preferably Rpb3-HA).

- in Figure 2, the effect on mature snR3 and snR99 should be quantified after normalization on the 5S.
- Figure 3B: the normalization of the data should be better explained. As the scale used between samples is different, the number of downstream reads for pab2 could be very significant.
- Figure 4: the mapping of phospho-CTD has already been performed by several teams in pombe and this should be properly referenced. In addition, there is no reason not to show full-length snoRNA in Figure 4B, which would facilitate the analysis.
- Figure 5D-G: it would be better to show the Fig S5B with normalized data, which clearly shows that the effect of dhp1 depletion is very weak. The non-normalized data is misleading.
- « data not shown » should be avoided

major issues

- many key conclusions of the paper rely on the use of conditional mutants, which is necessary as all the proteins analysed are encoded by essential genes. Contrary to thermosensitive mutants that can inactivate a protein within minutes, the use of the switch-off promoter is problematic for two reasons. First, it is well-documented that the strong nmt1 promoter that is used does not properly switch-off, which is why alternative versions were developed (several papers from Maundrell in the 90ies). Surprisingly, the switch-off of seb1, dhp1 and pcf11 is efficient on plate as concluded from cell death. However, no data is presented in liquid. Northern blots and western blots analyses MUST be presented to conclude that the proteins are indeed depleted in the time frame of the experiments. The details of the incubation times and thiamine concentration used are difficult to find. In the material and methods, it is stated that 12-15 hours were needed to switch-off the promoters at concentration of 60 uM thiamine (4 times the usual concentration). The 12-15 hours period represents at least 5 division cycles for pombe. How can the authors then conclude that the observed effects are direct? This problem may also explain the surprisingly weak effect of some depletion (see below).

The second problem is the fact that the nmt1 (when ON) is a very strong promoter and that the proteins are then overexpressed in the ON condition. It is very difficult to predict if this has an impact but Figure 1 is quite worrying: compare the peak of Pol II in Figure 1A in the « wt » (presumably the real wt) and the « wt » used for nmt-seb1 (presumably nmt-seb1 in the absence of thiamine). There is a shift of at least 50 bp.

There is no indication of the actual level of the proteins analysed before and after depletion, and compared to the wild type level.

In line with this, the author use the anchor-away system for nuclear depletion of Rna14 and Ysh1. Here the depletion is much more documented and convincing (Fig S2). Yet, the material and methods indicate that the depletion was performed for 4-15h. It is necessary to indicate the time of depletion for every experiments. Maintaining cells in rapamycin for more than 4 hours seems useless as read-through transcripts are obvious already after 4 hours. In budding yeast, much shorter times are usually used, which warrants that direct, primary effects are observed.

- another major issue for most experiments is that the depletion mutant is compared to a wild type strain rather than the same strain with promoter ON. In Figure 2, the depleted strains should be compared to the same strain (for example nmt-pcf11) grown in the absence of thiamine, which is never shown. It is also important to test the effect of thiamine itself by using the wt strain but only as an additional control. The same comment stands true for the rapamycin experiments.

- Figure 4 is purely correlative and does not bring any key information. The authors conclude that « Ser2 and Tyr1 CTD phosphorylation are functionally relevant to fission yeast 3' end processing ».

Yet, several teams are reported that these marks are dispensable in fission yeast, contrary to all the proteins tested. Whatever the mechanism, this shows that the classical view of the CTD recruiting PCSF is over-simplified. This should be discussed here. In addition, the peaks of Pcf11 and Seb1 shown in Figure 4C do not fully overlap the peak of S2P. Why is that?

Obviously, the occupancy of Pol II and termination factors in the available CTD mutants would help to clarify the issue and make the paper appealing to a broader audience.

Reviewer #1 (Remarks to the Author):

This is an important and novel conclusion and could justify further consideration of this article in Nature Communication. However, a few major concerns should be addressed before.

RESPONSE: We thank Reviewer #1 for his/her interest in the study and the valuable comments. The specific concerns/questions of the Reviewer are addressed below.

MAJOR POINTS

In figure 2 the authors show that depletion of Pcf11, Ysh1 and Rna14 leads to the detection of polyadenylated snoRNA precursors, very similar to the ones detected in *pab2Δ* cells. The increased detection of precursors is not expected in termination mutants. This is seemingly at odd with a simple model according to which these factors cleave the nascent RNA to generate the precursors and provide an entry point for Dhp1. It is unclear how cleavage would be generated in the absence of Ysh1. If the termination defect is due to a cleavage defect (as suggested by the implication of Ysh1) the production of snoRNA precursors is expected to decrease to the benefit of longer species, with presumably poorly defined 3' ends. But the opposite is apparently observed. The 3'-extended species shown in figure 2 could be primary termination products and generated independently from CPF-dependent cleavage. Still, why are they only visible when the CPF is depleted? Is the CPF involved in the maturation of snoRNA precursors?

RESPONSE: The reviewer raises an excellent point. Collectively, our studies on 3' end processing and transcription termination at snoRNA genes have identified three types of mutants: (i) one type showing transcription termination defects but no detectable RNA processing deficiencies (*dhp1*), (ii) one type showing RNA maturation defects but normal transcription termination (*pab2*; Lemay et al. 2010, *Mol. Cell*), and (iii) mutants that show defects in both transcription termination and snoRNA 3' end processing (*pcf11*, *ysh1*, *rna14*, and *seb1*). At present, our data suggest that in addition to promoting the endonucleolytic cleavage for Dhp1-dependent termination, the CPF also contributes to snoRNA 3' end maturation in fission yeast. We have now clarified this point in the Discussion of the revised manuscript (see p. 15-16, lines 30-34 and 1-8) as well as included this view in our revised model (see Fig .7D-7E and p. 26, 32-34).

Similarly, why the first PAS signal of snR99 – which is supposed to be the PAS used in wild type cells to induce termination – is strongly selected in Pcf11-depleted cells (Fig3B) ? Or is it always selected with the same efficiency and the RNA not processed in the absence of Pcf11 ?

RESPONSE: As expected, depletion of Pcf11 resulted in transcription termination defects as demonstrated by read-through RNAPII (Supplementary Fig. 6) and long read-through transcripts (Fig. 2F). In addition, depletion of Pcf11 also resulted in the accumulation of polyadenylated snoRNA precursors, as demonstrated by our RNase H (Fig. 2) and 3'READS (Fig. 3) data, together with reduced levels of mature snoRNAs (Fig. 2). As discussed in the aforementioned point, these data are consistent with the dual role of the CPF in transcription termination and RNA 3' processing (see p. 15-16).

The issue of processing versus termination could be addressed by coupling a processing

defect (maybe *pab2Δ*) for visualizing the precursors to the putative termination defect and showing that the precursors disappear when termination is affected.

RESPONSE: *As suggested by Reviewer #1, we generated a double *Pnmt-pcf11 pab2Δ* double mutant to get insights into the mechanism underlying the accumulation of polyadenylated *snoRNA* precursors in the CPF mutants. Comparison of the *Pnmt-**

pcf11 pab2Δ double mutant to the *Pnmt-pcf11* and *pab2Δ* single mutants revealed two main observations (see below). First, the accumulation of polyadenylated pre-*snoRNAs* in the double mutant is similar to the *Pnmt-pcf11* and *pab2Δ* single mutants. This non-additive phenotype is consistent with the view that the CPF contributes to *Pab2*-dependent 3' end maturation of *snoRNA* precursors (see model Fig. 7D-7E). In contrast, an additive effect was observed for the reduction of mature *snoRNA* levels in the double mutant as compared to the single mutants, which can be explained by the transcription termination defect detected in *Pcf11*-deficient cells (Supplementary Fig. 6), but not in the *pab2* mutant

(Lemay et al. 2010 Mol cell.). Collectively, the results obtained from the analysis of the *Pnmt-pcf11 pab2Δ* double mutant are consistent with a dual role of the CPF in transcription termination and *snoRNA* 3' end processing in fission yeast.

It is also unclear what is the origin of the long RT species shown in figure 2F. These could be evidence of readthrough transcription – as the authors suggest – but also additional processing sites that are only detected in mutants (akin to the 3'-extended species of figure 2B-E). It is essential that the authors clarify these points and provide a more direct evidence of transcription termination defects (i.e. polymerase detection) in *Pcf11* and *Rna14* cells.

RESPONSE: *As requested by the reviewer, the revised manuscript includes analyses of RNAPII occupancy in *pcf11* and *rna14* mutants, which show clear evidence of read-through transcription at *snoRNA*-coding genes (see Supplementary Fig. 6A-6B) as well as other long *ncRNA* genes (see Supplementary Fig. 6C-6E). Thus, the robust read-through transcription detected in CPF mutants (*ysh1*, *pcf11*, and *rna14*) is likely the cause of the long read-through RNA species seen in Fig. 2F, which are not detected in the *pab2* mutant (Fig. 2F, lane 8) because this mutant does not exhibit transcription termination defects (Lemay et al. 2010, Mol. Cell.). These new data are now described on p. 12 (lines 10-16) of the revised manuscript.*

The authors use “ncRNAs” throughout the paper while in reality they only show data concerning *snoRNAs* (and one *snRNA*). But many additional non-coding RNAs are also present in *S. pombe*: is transcription termination also affected at other classes of non-coding RNAs in CPF mutants? They should perform such an analysis before concluding that the function of the *S.cerevisiae* NNS complex is not conserved in *S.pombe* and that the

CPF terminates transcription of ncRNA genes. This might also help clarifying the question of processing vs termination raised above.

RESPONSE: *The current study focuses on transcription termination by RNA polymerase II. Accordingly, we have not investigated transcription termination at rRNA, tRNA, and 7SL (SRP) genes, which are produced by RNA polymerase I and III. The original manuscript primarily focused on snoRNA and snRNA genes, as these ncRNAs are well-studied targets of the NNS pathway in S. cerevisiae. As recommended by Reviewer #1, we have now examined other types of non-coding RNAs in fission yeast and found that they also show enrichment of CTD phosphorylation marks and binding of CPF factors similar to mRNA, snoRNA, and snRNA genes. Importantly, we show that transcription termination at ncRNA genes is dependent on CPF (Ysh1, Pcf11, and Rna14) and termination (Dhp1) factors. These new data are now presented in Fig. 7A-7C and Supplementary Fig. 6C-6E of the revised manuscript and described on p. 13 (lines 7-28) of the Result section. We thank the review for this excellent suggestion, which strengthens our study.*

Reviewer #2 (Remarks to the Author):

The paper by Larochelle et al makes a convincing case for a novel form of 3' end formation and termination of transcription at the 3' ends of non-coding RNAs in fission yeast. The data in support of the conclusions are thorough and of high quality. The conclusions for the most part are compelling and important because they establish a significant diversity in the mechanisms responsible for ncRNA 3' end formation with substantial differences between budding and fission yeast. I have only minor suggestions.

RESPONSE: *We would like to thank this reviewer for the constructive comments and enthusiasm for the study. This reviewer raised several excellent questions that are addressed below.*

1. It would be good to provide some explanation for the difference in termination sites between the two WT strains in Figs 1A, B.

RESPONSE: *We thank the reviewer for raising this point. Fig. 1A and 1B show ChIP-seq data using the same wild-type strain but grown in either rich (YES) or minimal (EMM) media. We have clarified this point in the legend to Figure 1.*

2. It is counter-intuitive that mutation of the cleavage-polyadenylation factors Pcf11, Ysh1, and Rna14 should lead to accumulation of polyadenylated snoRNA transcripts and the authors should provide an interpretation of this result to help the reader.

RESPONSE: *The reviewer raises an excellent point. Collectively, our studies on 3' end processing and transcription termination at snoRNA genes have identified three types of mutants: (i) one type showing transcription termination defects but no detectable RNA processing deficiencies (dhp1), (ii) one type showing RNA maturation defects but normal transcription termination (pab2; Lemay et al. 2010, Mol. Cell), and (iii) mutants that show defects in both transcription termination and snoRNA 3' end processing (pcf11, ysh1, rna14, and seb1). At present, our data suggest that in addition to promoting the endonucleolytic cleavage for Dhp1-dependent termination, the CPF also contributes to snoRNA 3' end maturation in*

fission yeast. We have now clarified this point in the Discussion of the revised manuscript (see p. 15-16, lines 30-34 and 1-8) as well as included this view in our revised model (see Fig .7D-7E and p. 26, 32-34).

3. In figure 3 do the cleavage sites in the chimeric snR99-rps2 and snR99-trx1 genes correspond to those used in the natural Rps2 and Trx1 mRNAs?

RESPONSE: Yes. We have now done 3' RACE analysis using total RNA from strains that expressed the snR99-rps2 and snR99-trx1 chimeric constructs and found that the chimeric snoRNAs use similar cleavage sites as those used for endogenous rps2 and trx1 mRNAs. This data is now presented in Supplementary Fig. 3C-3D and described on p. 9 (lines 18-19) of the revised manuscript.

4. The use of the term “rapid bursts of Ser5-P and Ser7-P” line 32 p. 9 is inappropriate. This is not a kinetic experiment.

RESPONSE: We agree and have changed the sentence to: “with peaking Ser5-P and Ser7-P signals at the 5' end...”. We thank the reviewer for making note of this incorrect statement.

5. The authors should acknowledge the reported localization of Rat1 on snoRNA genes in budding yeast (ref. 18) in common with the results reported here for fission yeast.

RESPONSE: We have now denoted this common observation in the discussion of the revised manuscript (see p. 15, line 15-19).

Reviewer #3 (Remarks to the Author):

We thank this reviewer for his/her comments and suggestions. The specific comments and questions of the reviewer are addressed below.

MAJOR ISSUES

- many key conclusions of the paper rely on the use of conditional mutants, which is necessary as all the proteins analysed are encoded by essential genes. Contrary to thermosensitive mutants that can inactivate a protein within minutes, the use of the switch-off promoter is problematic for two reasons. First, it is well-documented that the strong nmt1 promoter that is used does not properly switch-off, which is why alternative versions were developed (several papers from Maundrell in the 90ies). Surprisingly, the switch-off of seb1, dhp1 and pcf11 is efficient on plate as concluded from cell death. However, no data is presented in liquid. Northern blots and western blots analyses MUST be presented to conclude that the proteins are indeed depleted in the time frame of the experiments. The details of the incubation times and thiamine concentration used are difficult to find. In the material and methods, it is stated that 12-15 hours were needed to switch-off the promoters at concentration of 60 uM thiamine (4 times the usual concentration). The 12-15 hours period represents at least 5 division cycles for pombe. How can the authors then conclude that the observed effects are direct? This problem may also explain the surprisingly weak effect of some depletion (see below).

RESPONSE: First, it should be clear that the strong nmt1 promoter was NOT used in this study; only the weaker nmt41 and nmt81 versions of the nmt promoter were used. This was outlined in the original version of the manuscript in Supplementary

Table 3, which described the list of strains used in the study. We have now stressed this point in the result section of the revised manuscript (see p. 6, lines 20-24). Second, as requested by the reviewer, the revised manuscript now includes Western blot analysis of Dhp1, Seb1, and Pcf11 conditional strains showing robust depletion in the time frame of the described experiments (described in p. 6, lines 20-24 and shown in Supplementary Fig. 2A-2C of the revised manuscript). Third, 12h-15h incubation times with thiamine is commonly used with the nmt promoters and is certainly less than the 24h used by other groups that analyzed similar factors (see Wittmann S et al. 2017, Nature Communications 8:14861).

The second problem is the fact that the nmt1 (when ON) is a very strong promoter and that the proteins are then overexpressed in the ON condition. It is very difficult to predict if this has an impact but Figure 1 is quite worrying: compare the peak of Pol II in Figure 1A in the « wt » (presumably the real wt) and the « wt » used for nmt-seb1 (presumably nmt-seb1 in the absence of thiamine). There is a shift of at least 50 bp.

RESPONSE: The strong nmt1 promoter was NOT used here, as mentioned above. Importantly, we have addressed the overexpression concern of this reviewer by comparing the levels of Dhp1, Seb1, and Pcf11 proteins expressed from their endogenous promoter as opposed to the conditional nmt41/81 promoter. As shown in Supplementary Fig. 2A-2C of the revised manuscript, the levels of Dhp1, Seb1, and Pcf11 expressed from the nmt promoter in the absence of thiamine (non-repressing conditions) are similar to levels detected from their endogenous promoters (compare lanes 1 and 3). Furthermore, Fig. 1A and 1B show ChIP-seq data using the same wild-type control strain but grown in either rich (YES) or minimal (EMM) media. We have clarified this point in the legend to Figure 1.

There is no indication of the actual level of the proteins analysed before and after depletion, and compared to the wild type level. In line with this, the author use the anchor-away system for nuclear depletion of Rna14 and Ysh1. Here the depletion is much more documented and convincing (Fig S2). Yet, the material and methods indicate that the depletion was performed for 4-15h. It is necessary to indicate the time of depletion for every experiments. Maintaining cells in rapamycin for more than 4 hours seems useless as read-through transcripts are obvious already after 4 hours. In budding yeast, much shorter times are usually used, which warrants that direct, primary effects are observed.

RESPONSE: As mentioned above, we now provide Western blot analyses of Dhp1, Seb1, and Pcf11 before and after depletion, as well as comparison of their expression levels when expressed from their endogenous promoter as opposed to the nmt promoter (Supplementary Fig. 2A-2C). As recommended by the reviewer, we have now indicated the time of depletion used for each experiment in the legends to figures of the revised manuscript.

- another major issue for most experiments is that the depletion mutant is compare to a wild type strain rather than the same strain with promoter ON. In Figure 2, the depleted strains should be compared to the same strain (for example nmt-pcf11) grown in the absence of thiamine, which is never shown. It is also important to test the effect of thiamine itself by using the wt strain but only as an additional control. The same comment stands true for the rapamycin experiments.

RESPONSE: As requested by Reviewer #3, the revised manuscript now includes RNase H cleavage assays that compare the Pnmt-pcf11, Pnmt-seb1, and Pnmt-dhp1

strains with and without the addition of thiamine. This control experiment, which is now presented as Supplementary Fig. 2K of the revised manuscript, shows that the Pnmt conditional mutants are not leaky in non-repressing conditions (absence of thiamine; compare lanes 1, 3, 5, and 7). The rapamycin dependency of the ysh1-AA and rna14-AA conditional strains was demonstrated in the original version of our manuscript (see Supplementary Fig. 2H-2J of the revised manuscript), which showed a time-dependent increase in readthrough transcripts (and concomitant reduction in mature mRNA) after the addition of rapamycin.

- Figure 4 is purely correlative and does not bring any key information. The authors conclude that « Ser2 and Tyr1 CTD phosphorylation are functionally relevant to fission yeast 3' end processing ». Yet, several teams are reported that these marks are dispensable in fission yeast, contrary to all the proteins tested. Whatever the mechanism, this shows that the classical view of the CTD recruiting PCSF is over-simplified. This should be discussed here.

RESPONSE: We disagree. To our knowledge, the fact that Ser2 and Tyr1 CTD phosphorylation correlate with 3' end processing factors at snoRNA genes has never been reported. Given the reciprocal relationship between Ser2 phosphorylation and 3' end processing in S. cerevisiae and humans, the data presented in Figure 4 provide an additional line of evidence supporting co-transcriptional 3' end processing of snoRNA genes by the CPF in fission yeast, and is therefore of relevance. Yet, as noted by the reviewer, the absolute requirement of Ser2 CTD phosphorylation for RNA 3' end processing remains controversial in fission yeast. This point has now been raised in the revised manuscript (see p. 16, lines 25-30), as suggested by Reviewer #3.

Obviously, the occupancy of Pol II and termination factors in the available CTD mutants would help to clarify the issue and make the paper appealing to a broader audience.

RESPONSE: We agree, and this is ongoing. We prefer using full-length CTD mutants rather than the available minimal truncated versions. Nevertheless, this question extends beyond the scope of the current study.

MINOR COMMENTS

- the use of the word « non-coding RNA (ncRNA) » while referring to snoRNAs and snRNAs is confusing. They are many more ncRNA (about 1500) in fission yeast that do not belong to this category and whose transcription is similar to the coding RNAs. In addition, rRNAs and tRNAs are also obviously ncRNAs.

RESPONSE: The current study focuses on transcription termination by RNA polymerase II. Accordingly, we have not investigated transcription termination at rRNA, tRNA, and 7SL (SRP) genes, which are produced by RNA polymerase I and III. The original manuscript primarily focused on snoRNA and snRNA genes, as these ncRNAs are well-studied targets of the NNS pathway in S. cerevisiae. As recommended by Reviewer #1, we have now examined other types of non-coding RNAs in fission yeast and found that they also show enrichment of CTD phosphorylation marks and binding of CPF factors similar to mRNA, snoRNA, and snRNA genes. Importantly, we show that transcription termination at ncRNA genes is dependent on CPF (Ysh1, Pcf11, and Rna14) and termination (Dhp1) factors. These

new data are now presented in Fig. 7A-7C and Supplementary Fig. 6C-6E of the revised manuscript and described on p. 13 (lines 7-28) of the Result section.

- some metagene analyses are difficult to visualize due to the number of tracks and use of similar colours to distinguish them.

RESPONSE: We agree that some metagene plots are dense, but we think it is balanced by an easier comparison of the datasets. Before preparing the figures, we considerably discussed among ourselves on the best way to enable an intuitive presentation of the 25 different ChIP-Seq datasets showed in our study, and decided to consistently use a unique color for each of the targeted IP (for a total of 12 colors as distinct as possible), as well as use plain vs dashed/doted lines for the WT and mutant strains. We also ensured to combine colors in a way that facilitate their discrimination, and to limit to six the number of different datasets in a same panel. There is one notable exception to these rules that are Fig. 1A and 1B where the WT and mutant strains of NNS components are hard to distinguish from each other (all in orange), which was intentional to highlight the fact that they are actually all similar (much more than the seb1-mutant strain relative to its corresponding WT strain).

- in Figure 1, it would make sense to show the peak of S2P normalized on PolII (preferably Rpb3-HA).

RESPONSE: Figure 1 focuses on the binding of 3' end processing and termination factors (Rna14, Pcf11, Seb1, Ysh1, and Dhp1) at mRNA, snoRNA, and snRNA genes. The comparison between of Ser2-P RNAPII and 3' end processing factors is presented in Figure 4.

- in Figure 2, the effect on mature snR3 and snR99 should be quantified after normalization on the 5S.

RESPONSE: As requested by Reviewer #3, we have quantified the levels of mature snR3 and snR99 after normalization to the 5S rRNA. This information is now included in the Results section of the revised manuscript (see p. 7, lines 4-8).

- Figure 3B: the normalization of the data should be better explained. As the scale used between samples is different, the number of downstream reads for pab2 could be very significant.

RESPONSE: We would first like to thank Reviewer #3 for noting the omission of the mapping and normalization procedures of the 3'READS data; this was added in the Supplementary Methods of the "3'READS analysis" section of the revised Supplementary Information. It should be noted that during the revision process, we realized having used the non-normalized data file for the initial Fig. 3B and 3D (and Supplementary Fig. 3A-3B). The revised manuscript contains the normalized data showing almost identical results, and therefore not affecting any of our conclusions.

Considering that the data was normalized by Reads Per Million (RPM; such that the total signal over the whole-genome of each sample is equivalent), we could have used the same scale for all the samples in Fig. 3B to ease the visual representation of the 47-fold increase of normalized reads between the WT and the Pab2 mutant. Yet, this would have obscured the attention that the major site is the same between the WT strain and the pab2 mutant. Using different scales also allow appreciating the similar 4- and 7-fold increase in WT and Pab2 mutant, respectively, between the

major site and the one downstream (which would have been impossible to see using the same scale for all samples).

- Figure 4: there is no reason not to show full-length snoRNA in Figure 4B, which would facilitate the analysis.

RESPONSE: *Considering that the length of the snoRNA genes vary less than the length of protein-coding genes, we could have used a relative approach as shown below to represent the average ChIP signal at snoRNA genes. Yet, as Reviewer #3 can appreciate, the biological interpretation is exactly the same as using an absolute approach (as shown in Fig. 4B). The relative approach implies that genes are represented in a constant number of windows (consequently longer genes contain larger windows potentially aggregating signal not from the same part of different genes (e.g. a signal 600 bp downstream of the TSS would be contained in the 6th window of a 1kb gene divided in 10 windows, but in the 2nd window of a 3kb gene*

also divided in 10 windows)), therefore without interruption. The absolute approach implies that genes are represented in windows of constant length (consequently longer genes need more windows) and in practice we only show a define number of windows corresponding to the first and last parts of the genes, thus triggering an interruption. As the absolute approach is more appropriate for mRNA genes and considering that both approaches are equivalent for snoRNA genes, we decided to use a

uniform methodology for both coding and non-coding genes. In the revised manuscript, we nonetheless modified in Fig. 4B, 5E, 5G, and 6E (and the corresponding description in the Supplementary Methods) to increase the length of the snoRNA genes covered in the metagene plots.

- Figure 5D-G: it would be better to show the Fig S5B with normalized data, which clearly shows that the effect of dhp1 depletion is very weak. The non-normalized data is misleading.

RESPONSE: *In this study, we present both the non-normalized (Fig. 5F-5G) and RNAPII-normalized (Supplementary Fig. 5B) data. We believe that presenting the non-normalized data is important to get a comprehensive view of the changes seen for Ser2-P and Tyr1-P profiles in cells deficient of Dhp1. Specifically, the Ser2-P and Tyr1-P ChIP-seq data normalized to total RNAPII shown in Supplementary Fig. 5B shows that the levels of Dhp1 clearly affects the levels of Ser2 and Tyr1 CTD phosphorylation downstream of genes.*

- « data not shown » should be avoided

RESPONSE: *We agree and have made the correction.*

REVIEWERS' COMMENTS:

Reviewer #1 (Remarks to the Author):

The authors have responded to my criticisms and have modified the manuscript accordingly. I can now recommend publication of the manuscript in Nature Communication

Reviewer #3 (Remarks to the Author):

The revision has improved the manuscript a lot and the authors have made most important changes requested. I apologize for missing the fact that weaker versions of the thiamine-repressible promoter were used rather than nmt1. However, the issue remains but was addressed by showing the western blots. It is surprising that the nmt (ON) always results in identical expression level to the endogenous promoters. Maybe there is a post-transcriptional level of buffering control of these proteins.

I am not convinced by the response of the authors that "12h-15h incubation times with thiamine is commonly used with the nmt promoters and is certainly less than the 24h used by other groups that analyzed similar factors (see Wittmann S et al. 2017, Nature Communications 8:14861)". Claiming that other groups do worse is not an argument. In addition, though I agree that the repression is robust, there is still quite a few proteins left and to me, this is a serious problem considering the long incubation time (4 to 5 division cycles). On the other hand, I don't have any good alternative to propose beside using a ts allele.

Regarding the issue of the requirement of Ser2 CTD phosphorylation for RNA 3' end processing, it cannot be stated that this is "controversial" as it has never been tested (to my knowledge). The only fact is that a S2A mutant is viable in both fission yeast and budding yeast, contrary to RNA 3' end processing factors that are essential. The most likely explanation is that these complexes could be recruited through RNA binding as shown for elongation factors in the Battaglia et al. paper (eLIFE 2017). This should be discussed accordingly.

Reviewer #1 (Remarks to the Author):

The authors have responded to my criticisms and have modified the manuscript accordingly. I can now recommend publication of the manuscript in Nature Communication

RESPONSE: We thank the reviewer for taking the time to evaluate the revised version of our manuscript.

Reviewer #3 (Remarks to the Author):

The revision has improved the manuscript a lot and the authors have made most important changes requested. I apologize for missing the fact that weaker versions of the thiamine-repressible promoter were used rather than nmt1. However, the issue remains but was addressed by showing the western blots. It is surprising that the nmt (ON) always results in identical expression level to the endogenous promoters. Maybe there is a post-transcriptional level of buffering control of these proteins.

I am not convinced by the response of the authors that "12h-15h incubation times with thiamine is commonly used with the nmt promoters and is certainly less than the 24h used by other groups that analyzed similar factors (see Wittmann S et al. 2017, Nature Communications 8:14861)". Claiming that other groups do worse is not an argument. In addition, though I agree that the repression is robust, there is still quite a few proteins left and to me, this is a serious problem considering the long incubation time (4 to 5 division cycles). On the other hand, I don't have any good alternative to propose beside using a ts allele.

RESPONSE: We thank the reviewer for taking the time to evaluate the revised version of our manuscript. Quantification of Western blot data indicate that Dhp1, Pcf11, and Seb1 were depleted by more than 85% after thiamine-dependent repression of the nmt1 promoter, showing functional impact on 3' end processing and transcription termination relative to control cells treated with thiamine-supplemented media.

Regarding the issue of the requirement of Ser2 CTD phosphorylation for RNA 3' end processing, it cannot be stated that this is "controversial" as it has never been tested (to my knowledge). The only fact is that a S2A mutant is viable in both fission yeast and budding yeast, contrary to RNA 3' end processing factors that are essential. The most likely explanation is that these complexes could be recruited through RNA binding as shown for elongation factors in the Battaglia et al. paper (eLIFE 2017). This should be discussed accordingly.

RESPONSE: We thank the reviewer for the suggestion. Accordingly, we have included a reference to the Battaglia paper as an example of a redundant mechanism to recruit 3' end processing factors co-transcriptionally (see p. 16, lines 19-21).